# Coupling of ocean redox and animal evolution during the Ediacaran-Cambrian transition

Dan Wang[1,2], Hong-Fei Ling[2], Ulrich Struck[3,4], Xiang-Kun Zhu[1], Maoyan Zhu[5,6], Tianchen He[7], Ben Yang[8], Antonia Gamper[3,4] & Graham A. Shields [9]

The late Ediacaran to early Cambrian interval witnessed extraordinary radiations of metazoan life. The role of the physical environment in this biological revolution, such as changes to oxygen levels and nutrient availability, has been the focus of longstanding debate. Seemingly contradictory data from geochemical redox proxies help to fuel this controversy. As an essential nutrient, nitrogen can help to resolve this impasse by establishing linkages between nutrient supply, ocean redox, and biological changes. Here we present a comprehensive N-isotope dataset from the Yangtze Basin that reveals remarkable coupling between $\delta^{15}N$, $\delta^{13}C$, and evolutionary events from circa 551 to 515 Ma. The results indicate that increased fixed nitrogen supply may have facilitated episodic animal radiations by reinforcing ocean oxygenation, and restricting anoxia to near, or even at the sediment–water interface. Conversely, sporadic ocean anoxic events interrupted ocean oxygenation, and may have led to extinctions of the Ediacaran biota and small shelly animals.

[1] MNR Key Laboratory of Isotope Geology, MNR Key Laboratory of Deep-Earth Dynamics, Institute of Geology, Chinese Academy of Geological Sciences, Beijing 100037, China. [2] State Key Laboratory for Mineral Deposits Research, School of Earth Sciences and Engineering, Nanjing University, Nanjing 210023, China. [3] Museum für Naturkunde, Leibniz Institute for Evolution and Biodiversity Science, 10115 Berlin, Germany. [4] Department of Earth Sciences, Freie Universität Berlin, Malteserstrasse 74-100, Haus D, 12249 Berlin, Germany. [5] State Key Laboratory of Palaeobiology and Stratigraphy, Center for Excellence in Life and Paleoenvironment, Nanjing Institute of Geology and Palaeontology, Chinese Academy of Sciences, 210008 Nanjing, China. [6] College of Earth Sciences, University of Chinese Academy of Sciences, 100049 Beijing, China. [7] School of Earth and Environment, University of Leeds, Leeds LS2 9JT, UK. [8] MNR Key Laboratory of Stratigraphy and Palaeontology, Institute of Geology, Chinese Academy of Geological Sciences, 100037 Beijing, China. [9] Department of Earth Sciences, University College London, Gower Street, London WC1E 6BT, UK. Correspondence and requests for materials should be addressed to D.W. (email: njuwangdan@gmail.com) or to H.-F.L. (email: hfling@nju.edu.cn)

Soft-bodied macroscopic multicellular Ediacara organisms, including animals, appeared abruptly in the fossil records during the late Ediacaran Period[1], and were then replaced by bilaterian and biomineralized animals in ecosystems, which approached modern levels of complexity by the early Cambrian[2]. The underlying causes for biodiversification and bioradiations during the Ediacaran and early Cambrian are still controversial[3]. The key innovations in complex animals during the Ediacaran–Cambrian transition interval were skeletonization[4], motility[5], predation[6], large body size, and complex food webs[1,2]. While some simple animals, such as soft-bodied sponges, which may have their origins in the Tonian[7], could have had low oxygen requirements (0.5–4.0% of present-day levels)[8], macroscopic animals of the Ediacaran–Cambrian transition likely required oxygen levels above the physiological thresholds for those energetically expensive behaviors[9,10]. A gradual rise in oceanic oxygen levels has been proposed based on substantial geochemical evidence, e.g., the high abundance and isotope values of redox-sensitive trace metals (Mo and U) in euxinic sediments, and indeed seems to occur contemporaneously with animal evolution[11–14]. However, iron speciation data also suggest a high degree of redox heterogeneity, and possibly even dominantly anoxic conditions with scant measurable change in oxygenation through the Ediacaran to Cambrian interval[15–17]. A new model for ocean redox is, therefore, required to resolve these seemingly contradictory datasets, in particular where a direct relationship between animal radiation and extinction events can be established.

The N isotopic composition of marine sediments has great potential to provide clues about ocean redox structure and its direct links with bio-evolution[18]. The marine nitrogen cycle is driven by metabolic pathways that exhibit differing isotopic fractionations[19]. Nitrogen enters the oceanic N cycle mainly through the fixation of atmospheric $N_2$ ($N_2$-fixation) by autotrophs using Mo-nitrogenases with minor isotopic fractionation ($\leq 2‰$)[19]. This process requires high energy to break the molecular nitrogen bond, and can only be performed by some bacteria and archaea[20]. Denitrification and anaerobic ammonia oxidation (i.e., anammox) release nitrogen back to the atmosphere as nitrous oxide ($N_2O$) or $N_2$, and are the largest sinks of nitrogen

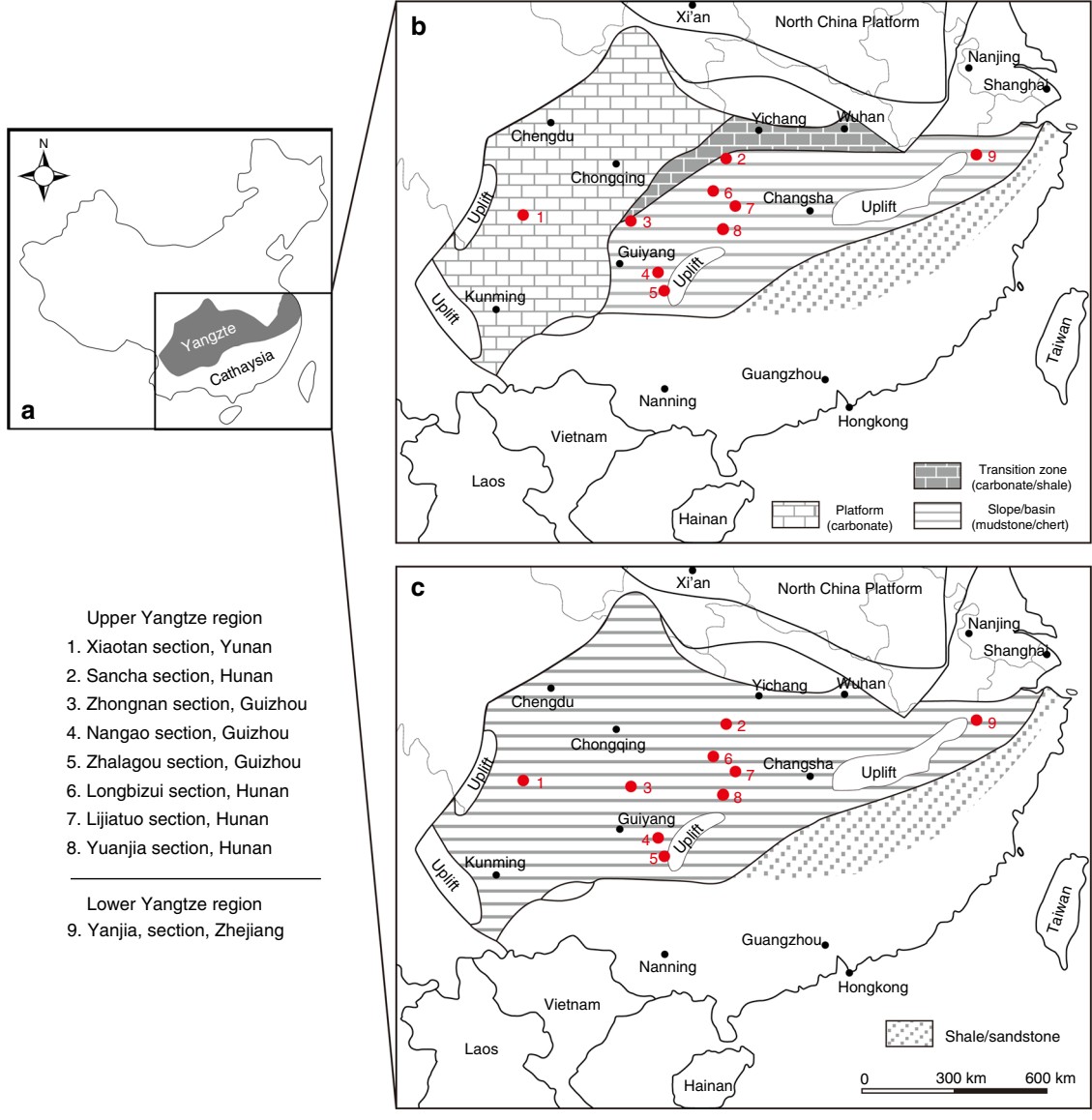

**Upper Yangtze region**

1. Xiaotan section, Yunan
2. Sancha section, Hunan
3. Zhongnan section, Guizhou
4. Nangao section, Guizhou
5. Zhalagou section, Guizhou
6. Longbizui section, Hunan
7. Lijiatuo section, Hunan
8. Yuanjia section, Hunan

**Lower Yangtze region**

9. Yanjia, section, Zhejiang

**Fig. 1** Tectonic setting and simplified geological map of the Yangtze block during the late Ediacaran–early Cambrian. **a** Tectonic setting of South China. **b** Simplified geological map of the Yangtze block during the late Ediacaran to Cambrian Fortunian Stage, and **c** during Cambrian Stage 2 (modified after ref. [24])

from the ocean[19]. These processes only occur under oxygen-deficient conditions, with significant fractionations of ~ 20–30‰ in the water column, but with negligible fractionation in the sediments[19,21]. The balance between nitrogen fixation and denitrification/anammox ultimately governs the fixed inorganic nitrogen reservoir in the ocean and its isotopic composition[19]. As it is a principal nutrient for life, nitrogen, by way of its redox sensitivity is considered to be an important control on the growth of eukaryotic producers and primary productivity[22]. However, nitrogen abundance and speciation, as well as its relationship with ocean redox state, productivity and animal

evolution across the Ediacaran–Cambrian transition, remain poorly understood.

Late Ediacaran to Cambrian successions are both fossiliferous and well developed along the Yangtze continental margin, South China, providing an excellent opportunity to conduct a comprehensive study of nitrogen isotopes (Fig. 1). This study presents a comprehensive curve of new and published $\delta^{15}N$ data from nine sections representing different depositional facies across South China. Our results demonstrate regional redox fluctuations against the background of a generally well-oxygenated open ocean. We find that ocean redox dynamics were closely coupled

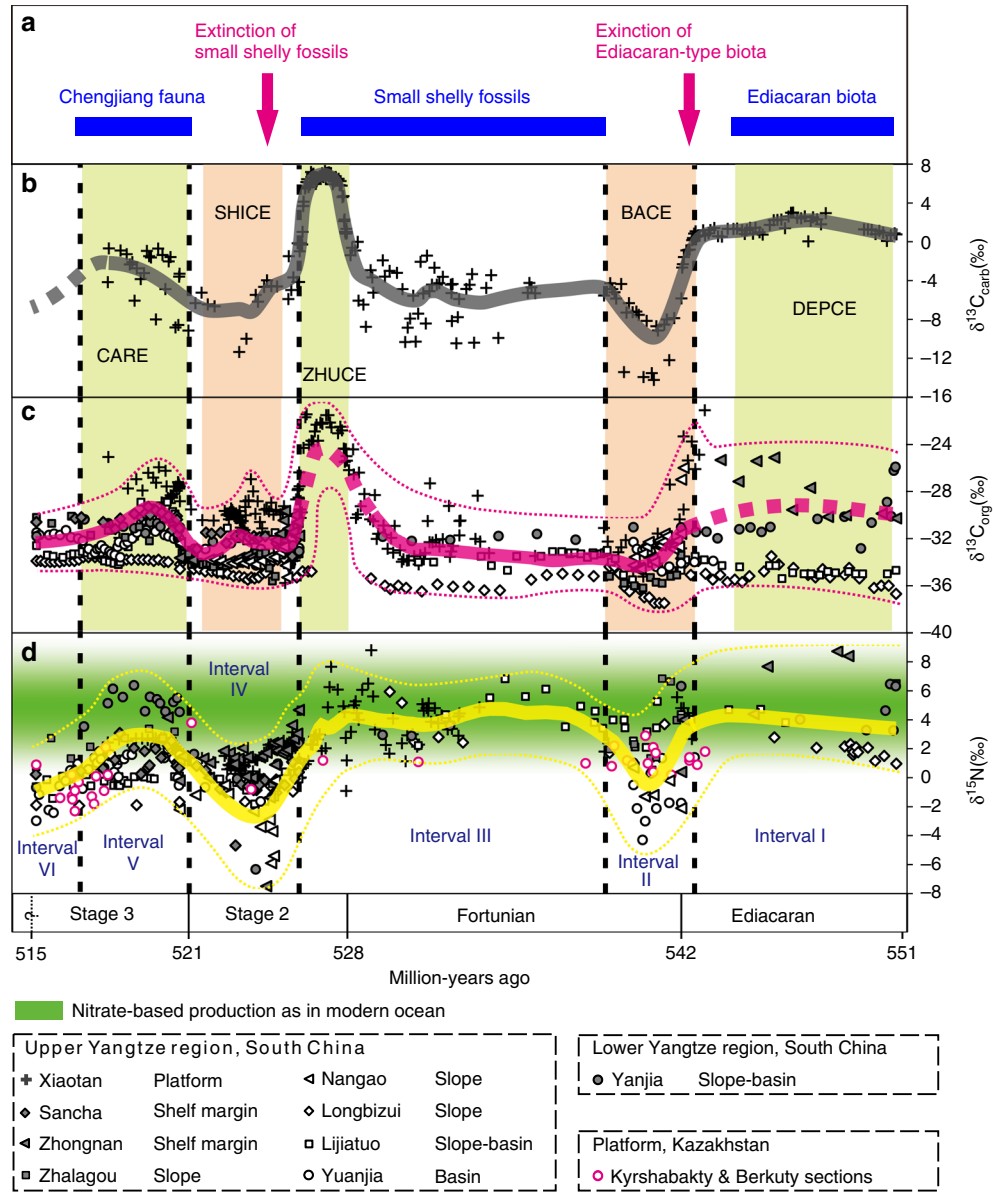

**Fig. 2** Compilation of $\delta^{13}C_{carb}$, $\delta^{13}C_{org}$, and $\delta^{15}N$ data with evolutionary events during the late Ediacaran–early Cambrian. **a** Key bioevents during the late Ediacaran–early Cambrian proposed by ref. [31]. **b** $\delta^{13}C_{carb}$ profile from the Xiaotan section (Yunnan) on the Yangtze platform. $\delta^{13}C_{carb}$ data are from refs. [67,68]. DEPCE: DEngying Positive Carbon isotope Excursion, BACE: Basal Cambrian Carbon isotope Excursion, ZHUCE: Zhujiaqing Carbon isotope Excursion, SHICE: Shiyantou Carbon isotope Excursion, CARE: Cambrian Arthropod Radiation isotope Excursion[31]. **c** Compiled $\delta^{13}C_{org}$ curve for nine sections in various depositional facies on the Yangtze block. [13]C_org data are from this study and refs. [26,28–30,69–71]. **d** Compiled $\delta^{15}N$ profile for nine sections in various depositional facies on the Yangtze block and two sections in the Malyi Karatau area of the southern Kazakhstan. $\delta^{15}N$ data are from this study and refs. [26–30,63]. The mean long-term trends of $\delta^{13}C_{carb}$, $\delta^{13}C_{org}$, and $\delta^{15}N$ data are given by LOWESS curves (bold lines) over the range of 10%. The positive and negative $\delta^{15}N$ excursions are statistically significant, as evidenced by Student's $t$-test ($p$-values < $10^{-4}$, see Supplementary Note 4 and Supplementary Fig. 6). The green area indicates that marine production by primary producers was based on nitrate as the dominant nutrient, as in modern oceans, with $\delta^{15}N$ values mainly ranging from +2‰ to +8‰

with key bio-events during the Ediacaran–Cambrian transition. We also suggest that redox fluctuations were probably of global significance, and linked to changes in nutrient abundances on geological time scales.

## Results

**Geological setting and stratigraphy.** The Yangtze block gradually evolved from a rift basin to a passive continental margin during the Proterozoic–Cambrian transition, with a southwest facing connection to the open ocean[23]. Late Ediacaran to early Cambrian sedimentary environments on the Yangtze block comprise proximal platform to distal basin realms, separated by a transitional slope setting (Fig. 1)[24]. During this time interval, carbonate predominantly deposited over the shallow-water platform, before migrating basinward into the black muds (i.e., cherts and shales) of the deeper marine environment[25] (Fig. 1b). During the early Cambrian, black shale deposition expanded across the entire Yangtze block in response to a major transgression (Fig. 1c). The basal part of these black shales is characterized by phosphorite nodules and Ni–Mo sulfide layers that can be used as correlation markers across the Yangtze region[25]. Nitrogen isotopic data were obtained from nine successions that represent shallow to deep marine environments (Fig. 1), including Xiaotan[26], Zhongnan[27], Sancha[28], Nangao[27], Zhalagou[29], Longbizui[30], Lijiatuo[30], and Yuanjia[28] sections in the upper Yangtze region, as well as Yanjia section (see Supplementary Note 1 and Supplementary Fig. 1 for lithological description) in the lower Yangtze region. The $\delta^{15}N$ data span the entire terminal Ediacaran to Cambrian Stage 3 (~551–515 Ma, see Supplementary Data 1). The stratigraphic correlation between deep and shallow marine facies was achieved using marker beds, reliable fossil records, precise radiometric dating, and carbon isotope ($\delta^{13}C_{carb}$ and $\delta^{13}C_{org}$) curves (see Supplementary Note 2 and Supplementary Figs. 3 and 4 for detailed descriptions). All nitrogen and carbon isotopic data from these nine sections are plotted in Fig. 2. Our generalized nitrogen isotope curve is broadly covariant with carbon isotope ($\delta^{13}C_{carb}$ and $\delta^{13}C_{org}$) records, and all three records reveal a striking relationship with known radiations and extinctions of early animals[31].

**Nitrogen isotopes.** All $\delta^{15}N$ curves from the different lithofacies show similar trends, although the absolute values are different for different sedimentary settings. In general, during the late Ediacaran (~551–542 Ma, interval I), the $\delta^{15}N$ values range between +1‰ and +9‰, approaching typical values for modern sediments; however, the mean value of +3.9‰ is lower than the modern mean value for $\delta^{15}N_{sediments}$ of +6.7‰[32]. During the Ediacaran–Cambrian transitional period (~542–539 Ma, interval II), some $\delta^{15}N$ data exhibit a negative shift toward low values around –4‰ to 0‰. Subsequently, from Cambrian Fortunian Stage 1 to early Stage 2 (~539–526 Ma, interval III), $\delta^{15}N$ values vary between +1‰ and +9‰, and show no significant differences from those of the late Ediacaran, with a similar mean value of +3.6‰. During Cambrian Stage 2 (~526–521 Ma, interval IV), a large negative shift occurs simultaneously at all sections with some very low values below –4‰ in the Yanjia, Nangao, Zhongnan, and Sancha sections. $\delta^{15}N$ values then increase at most sections within Cambrian Stage 3 (~521–518 Ma, interval V), although to different extents (averaging +1.8‰). During this fifth interval, $\delta^{15}N$ values from the Longbizui and Lijiatuo sections are the lowest, at around –2‰ to 0‰, i.e., close to the $\delta^{15}N$ value of atmospheric N[2][19], while $\delta^{15}N$ values at other sections lie between approximately +2‰ to +5.5‰. During the late Cambrian Stage 3 (~518–515 Ma, interval VI), $\delta^{15}N$ values decrease again to between –3‰ and 0‰.

## Discussion

As shown in Fig. 2, nitrogen isotopic variations seem to show hitherto unsuspectedly strong coupling with major bio-events across the Ediacaran–Cambrian transition. The times of relatively high $\delta^{15}N$ values are coincident with radiations of Ediacaran-type, small shelly-type, and Chengjiang-type biota during the intervals I, III, and V, respectively, which are also characterized by positive carbon isotope excursions. Conversely, shifts to negative $\delta^{15}N$ and $\delta^{13}C$ values occur synchronously at the intervals II, IV, and VI, whereby the first two are associated with the demise of Ediacaran fauna and small shelly animals, respectively[31]. Our results not only demonstrate significant redox fluctuations across the Ediacaran–Cambrian transition, but also reveal a direct correlation among nutrient availability, paleo-environments and bio-events.

Many Ediacaran and early Cambrian animal metabolisms likely required relatively high oxygen levels, although the exact thresholds have not yet been determined[9,10]. However, iron species data suggest that seawater was dominantly anoxic at depth around the Ediacaran–Cambrian transition[16,33], which seems at odds with the radiation of motile, benthic animals at this time. Our high $\delta^{15}N$ values for intervals I, III, and V, mainly ranging from +1 to +9‰ (except those in interval V of the Longbizui and Lijiatuo sections, Fig. 2d), are close to modern sediment values (+1 to +15‰)[32], reflecting a nitrogen cycle that approximates that of the modern ocean (Fig. 3a). These results imply that a large $NO_3^-$ reservoir was able to build up in well-oxygenated seawater during these intervals[34]. This conclusion is consistent with the increases in oxygen levels evidenced by redox-sensitive trace metal abundances and isotopes[11,13,14]. However, nitrate would be unlikely to accumulate in a ferruginous or sulfidic ocean, as it would have been reduced by $Fe^{2+}$ or $S^{2-}$ (ref. [34]). High $\delta^{15}N$ values during intervals I, III, and V therefore suggest a redox structure, in which anoxia, as recorded by iron species data[16,33], would have been restricted to near, or even at the sediment–water interface[34] (Fig. 3a), resolving the apparent contradictory implications of different redox proxy datasets. Pulses of oxygenation in deep waters could therefore have occurred during time intervals I, III, and V, removing physiological barriers to biological innovations among benthic Ediacaran-type and Cambrian-type animals (Fig. 2a).

As major diazotrophs, planktonic N-fixing cyanobacteria have been suggested to radiate into the open ocean during the late Neoproterozoic, substantially changing nitrogen supply in seawater[35,36], in response to a putative rise in phosphorus availability[37]. Increasing seawater phosphate could have been due to enhanced weathering, following major continental reorganisation and Cryogenian deglaciation[38], an elevated supply of oxidants for organic remineralization[39], and/or decreased scavenging through co-precipitation of Fe and P[40]. Enhanced supply of these bio-limiting nutrients (phosphorus and nitrogen) therefore modified the ecological structure of primary producers, likely facilitating a shift in phytoplankton composition from small-celled, cyanobacterial picoplankton toward large-celled, planktonic algae in the Neoproterozoic as evidenced from biomarker and molecular clock data[41,42]. Most importantly, a stable reservoir of nitrate existed in the substantially oxygenated ocean of the late Ediacaran–early Cambrian, as revealed by our $\delta^{15}N$ data for the time intervals I, III, and V. Large-celled, eukaryotic phytoplankton preferentially assimilate nitrate over ammonia[43], and so were able to thrive under nitrate-replete conditions, as diffusion limitation of nutrient uptake increases with cell size[44]. Thus, we propose that one initial driver for the flourishing of large-celled, eukaryotic phytoplankton may have been an increased nutrient-N (especially nitrate) supply. The trend toward more abundant eukaryotic phytoplankton created a more efficient biological

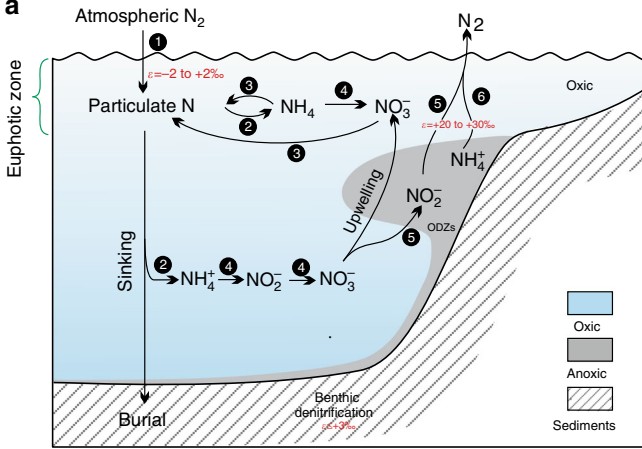

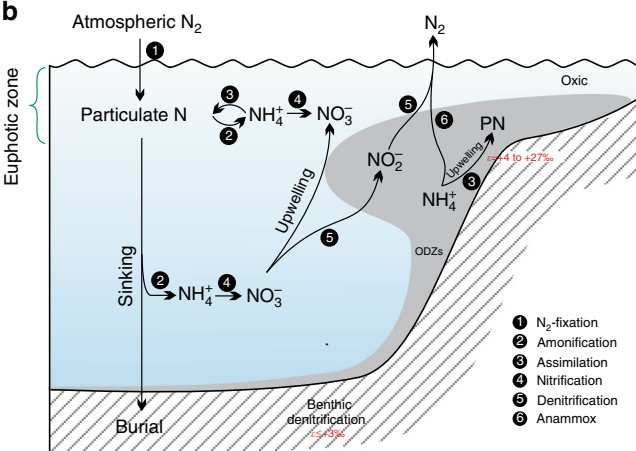

**Fig. 3** Schematic diagram of the biogeochemical nitrogen cycle for different time intervals during the late Ediacaran–early Cambrian in South China. **a** Time intervals I, III, and V. Modern-like nitrogen cycle operated in a considerably oxygenated ocean. Partial denitrification and/or anammox dominated nitrogen cycle in local oxygen-deficient zones. ODZs: Oxygen-Deficient Zones. **b** Time intervals II, IV, and VI. Anoxic events occurred against the background of an oxygenated open ocean. $NH_4^+$ assimilation or $N_2$-fixation dominated nitrogen cycle in expanded oxygen-deficient zones. PN: Particulate N. Nitrogen isotope fractionation ($\varepsilon = \delta_{reactant} - \delta_{product}$) is from refs. [18,19,21]

pump[41,45], which may have contributed to the positive $\delta^{13}C_{carb}$ and $\delta^{13}C_{org}$ excursions in time intervals I, III, and V (Fig. 2b, c). As a consequence, oxygenation of deep waters could occur in intervals I, III, and V, removing physiological barriers for bioradiations of benthic Ediacaran-type and Cambrian-type animals. Furthermore, the bloom of large-celled, eukaryotic phytoplankton also provided more complex food webs with more energy and carbon flowing to animals occupying higher trophic levels[37,41].

The feeding activity of animals, such as filtering and grazing, was suggested to contribute to the evolution of large-celled, eukaryotic phytoplankton as an important selective pressure, and in return the rise of eukaryotic phytoplankton led to a stepwise increase in export production, and hence reinforced marine oxygenation to sustain motile and macroscopic animals[45,46]. This may lead to a chicken-and-egg question, as the initial trigger for the positive feedback described above has still been in debate[3,47]. The earliest fossil record for suspension-feeding mesozooplankton was not reported to appear until the early Cambrian (510–515 Ma)[48]. Filter-feeding sponges evolved earlier, perhaps already in the Tonian Period when oxygen levels in seawater had

not yet significantly increased[7]. Sponges probably fed on free organic matter and picoplankton, which would have placed a selective pressure favoring large-sized eukaryotic phytoplankton[46]. However, top-down ecological pressure may not have been enough because the bloom of large eukaryotic phytoplankton still require a favorable environment with sufficient nutrient availability (e.g., nitrate and phosphorus) as a prerequisite. Therefore, even though it may be difficult to determine the causal directionality between ocean oxygenation and the origin of animals[3], our data could demonstrate that marine nutrient availability probably exerted a major control on both ocean oxygenation and macroevolution.

The marine environment in the late Ediacaran–early Cambrian was not as stable as the modern one. As shown in Fig. 2, $\delta^{15}N$ values at all nine sections synchronously shift toward $\delta^{15}N$ of ~0‰ with some values below –4‰ in intervals II, IV, and VI. The very low $\delta^{15}N$ values (<–2‰) could be interpreted as $N_2$-fixation using Fe-nitrogenases with a large isotopic fractionation (~8‰)[49]. However, Fe-nitrogenases were generally expressed only when Mo availability in seawater is extremely low[49], which is not the case in the Ediaracan–early Cambrian with high Mo concentrations (>100 ppm) recorded in the sediments[11,13,14]. Alternatively, these low $\delta^{15}N$ values (<–2‰) could result from partial $NH_4^+$-assimilation by living organisms with a large isotopic fractionation effect[50]. Notably, low $\delta^{15}N$ values below –2‰ rarely occurred in earlier Proterozoic[18], implying that marine nitrogen reservoir during the late Ediacanran–early Cambrian may be qualitatively different from the earlier times, and a larger $NH_4^+$ reservoir has been built up in the anoxic waters in intervals II, IV, and VI (Fig. 3b). These results further reflect the shoaling of the chemocline into the surface mixed layer, where $N_2$ and recycled $NH_4^+$ displaced $NO_3^-$ as the dominant nitrogen source during time intervals II, IV, and VI (Fig. 3b). Three negative carbon isotopic excursions correspond well to times of photic-zone anoxia (Fig. 2b, c), and this could be explained by the episodic expansion of anoxic waters with $^{13}C$-depleted inorganic carbon from the degradation of sinking organic matter[28]. The spread of anoxic waters containing toxic $H_2S$, as evidenced by iron speciation data[33,51,52], may have led to the extinction of Ediacaran-type and some small shelly-type animals (Fig. 2a), in a similar way that oceanic anoxic events (OAEs) can be related to most Phanerozoic extinctions[53,54].

The reasons for the occurrence of sporadic anoxic events along continental shelves are complex, but could plausibly be related to nutrient availability. A large fixed nitrogen reservoir has been built up in seawater, making nitrogen no long a limiting element during the late Ediacanran–early Cambrian. The bio-limiting nutrient phosphorus probably kept being efficiently recycled from organic matter and iron (hydr)oxides during the Ediacaran–Cambrian transition, as the seafloor remained largely anoxic due to its being covered by microbial mats[55,56]. The high nutrient availability likely fueled primary productivity, whereas resultant organic matter may only be partially buried in the sediments and grazed by predators (early animals). The remaining organic matter would have accumulated in the seawater, consuming oxygen and leading to a period of expansion of anoxia along the continental shelves during intervals II, IV, and VI, which in turn boosted nutrient (phosphorus) regeneration in a process of positive feedback[56]. The key to this positive feedback was the transfer of recycled nutrients from deep waters to the surface ocean, which would be closely related to ocean circulation[56]. A recent study of CIA (chemical index of alteration) data from South China proposed that a relatively cold and arid climate during Cambrian Stage 2 (interval IV) could have stimulated ocean circulation, encouraging upwelling during the extensive marine transgression[57]. It is therefore plausible that climate

change may have been an external force promoting episodic anoxia during the Ediacaran–early Cambrian period that bridged the transition from $O_2$-deficient oceans toward widely oxic oceans. However, CIA data from South China are lacking at the Ediacaran–Cambrian boundary (interval II), and their temporal correlation with our $\delta^{15}N$ data in Cambrian Stage 3 (intervals V and VI) are uncertain[57]. Therefore, more detailed correlation between climate change and redox fluctuations awaits future study.

Although phosphorus was more efficiently recycled from sediments under continuously anoxic conditions, increases in other P sinks, such as authigenic calcium phosphate, would eventually balance out sources and sinks[56]. Nitrogen enrichment in seawater, as revealed by the very low $\delta^{15}N$ values in intervals II, IV, and VI, would also be balanced through enhanced denitrification (and/or anammox) in $O_2$-deficient environments[19]. On the other hand, enhanced marine productivity and burial of organic matter would lead to increasing oxygen levels in the ocean and atmosphere, eventually counteracting ocean anoxia[58]. Thus, on long-term scales (>1 Myr), negative feedbacks would eventually be established, suppressing runaway positive feedbacks. This mechanism may explain the shifts from anoxic events to ocean oxygenation during the late Ediacaran–early Cambrian. In summary, seawater nutrient availability was able to regulate primary productivity and ocean redox over geological time (>1 Myr)[58], thereby creating a favorable environment for the evolution of metazoans.

The average $\delta^{15}N$ value of interval V is relatively lower compared to those of intervals I and III (Fig. 2d). We do not exclude the possibility that some of the low $\delta^{15}N$ values in interval V may reflect local signatures in oxygen-depleted zones where $N_2$-fixation dominated the nitrogen cycle (–2‰ to 0‰ in the Longbizui and Lijiatuo sections, Fig. 2d), as sea level may have dropped following the extensive transgression during Cambrian Stage 2 (interval IV)[28,51]. However, it is more likely that the lower $\delta^{15}N$ values represent global ocean deoxygenation in interval V relative to intervals I and III, as proposed previously[59,60]. The early Cambrian experienced significant "substrate revolution", and bioturbation significantly increased in interval V[61]. Increased bioturbation could have allowed more oxygen to penetrate into the sediments, enhancing nutrient (phosphorus) burial in the sediments[59]. The size of the global oxygen reservoir in interval V may therefore be smaller than during intervals I and III, owing to the decline in nutrient availability and productivity. Thus, life and environments co-evolved through the Ediacaran–Cambrian transition, eventually reaching a more stable Earth system, but approaching modern oxygen levels only later in the Phanerozoic[62].

An important issue arising is whether our findings are of global significance. To constrain this issue, $\delta^{15}N$ values from the Malyi Karatau area of southern Kazakhstan, so far the only known N-isotope data from the Ediacaran–Cambrian transition outside South China[63], are also plotted in Fig. 2 (see Supplementary Note 3 and Supplementary Fig. 5 for the stratigraphic correlation, and the $\delta^{15}N$ data in Supplementary Data 1). The $\delta^{15}N$ values are indeed consistent with the $\delta^{15}N$ curve from South China, but no data are available for interval I and data are too sparse for intervals III and IV from southern Kazakhstan to make a detailed comparison. Three anoxic events were identified in the Kazakhstan Basin, as with South China, and are evidenced from low $\delta^{15}N$ values, averaging ~1.3‰ in interval II, –0.8‰ in interval IV, and ~0.2‰ to –2.3‰ across time intervals V–VI (Fig. 2d). Although linkages between animal extinctions and regional anoxic events await systematic analysis and further study here and elsewhere, it is possible that dynamically changing and heterogeneous redox conditions on continental shelves were not

singular events of the Yangtze region, but also of global significance.

Carbon isotopic variations normally represent global changes in the ocean, and thus can be used for global stratigraphic correlation. The generalized $\delta^{15}N$ curve co-varies with carbon isotope trends (both $\delta^{13}C_{carb}$ and $\delta^{13}C_{org}$), further implying globally dynamic redox fluctuations across the Ediacaran–Cambrian transition (Fig. 2). High $\delta^{15}N$ values during time intervals I, III, and V imply the existence of a globally oxygenated open ocean, with anoxic conditions restricted to near, or even at the sediment–water interface. Under such conditions, fixed nitrogen could simply be replenished from the open ocean via surface currents into the Yangtze Basin. The increase in nutrient-N availability would have promoted the biological pump by fueling the productivity of large-celled, eukaryotic phytoplankton[35,41], thereby initiating a positive feedback toward further ocean oxygenation in step with early animal bio-radiations. However, sporadic anoxia, as evidenced by low $\delta^{15}N$ values in intervals II, IV, and VI, seem likely to have been regional events along local continental margins. This interpretation is supported by the elevated abundances and isotopic values of redox-sensitive metal elements in approximately contemporaneous anoxic sediments, indicating oxygenation of the oceans on a broader scale[13,28,33]. Such regional anoxic events may have been causally related to animal extinctions and occurred during times of excess oxygen consumption and/or insufficient oxygen production, which could be eventually regulated by nutrient availability. Extinctions of Ediacaran-type animals and the decline of small shelly animals, and their coeval negative carbon isotopic excursions, were also reported from other continents such as Siberia, Mongolia, and Morocco[1,64,65]. This further suggests that regional anoxia on continental shelves was not a singular event of the Yangtze region, but of global significance. It seems therefore conceivable that our local $\delta^{15}N$ data reflect wider, even global environmental changes.

In summary, our $\delta^{15}N$ data reveal an emerging picture that pulses of oxygenation of the deep oceans occurred during late Ediacaran–early Cambrian time, and was punctuated by regional anoxic events along continental margins during which the redox chemocline shoaled into the surface mixed layer. These redox fluctuations were probably of global significance and closely associated with changes in nutrient abundances on geological time scales. We further propose that increased nutrient (such as nitrogen) availability could have exerted an important control on both oxygenation and macroevolution during the late Ediacaran–early Cambrian, by boosting the primary productivity of large-celled, eukaryotic phytoplankton in the ocean. The sporadic anoxic events along continental margins may have been closely related to extinctions of Ediacaran-type and some early Cambrian-type animals.

## Methods

**Nitrogen isotope analysis.** Fresh samples were cut into pieces of ~3 cm, avoiding calcitic or siliceous veins where present, and then ground into a homogeneous powder. Approximately 100–200 mg of bulk sample powder was weighed in tin capsules, and measured for N isotopes with a Thermo Finnigan Elemental Delta V Mass Spectrometer coupled with a Thermo 1112 Flash Elemental Analyzer via a ThermoConfloIII Interface. Measurements were conducted in the Stable isotope laboratory of "Museum für Naturkunde" in Berlin. Nitrogen isotope delta values ($\delta^{15}N$) are reported relative to atmospheric air and calculated using a Peptone house-standard with $\delta^{15}N$ value of +7.6‰ (Supplementary Data 2). Replicate analyses of the samples yielded a precision better than ±0.3‰. The reliability of nitrogen isotope analytical performance has been proven previously in extensive "Round Robin" comparative testing[66].

**Organic carbon isotope analysis.** About 50–100 mg of each ground sample was weighed and leached in 2 M hydrochloric acid overnight to completely remove carbonate. Residues were washed with distilled water three times to remove

chlorides and dried in a drying oven at 40 °C overnight. Certain amounts of the decalcified sample powder were then wrapped into tin capsules for organic carbon isotope analysis. Measurements were conducted at the State Key Laboratory for Mineral Deposits Research, Nanjing University, using a Thermo EA-ConFloIV-MAT 253 isotope ratio mass spectrometer. Carbon isotope values ($\delta^{13}C_{org}$) are reported in per mil relative to the international V-PDB standard value. Replicate analyses of the samples yielded a precision better than ±0.3‰. Organic isotopic data enabled stratigraphic correlations, assisted by reliable fossil records and available isotope geochronology (see Supplementary Data 2, Supplementary Note 2 and Supplementary Figs. 3 and 4).

**Total nitrogen (TN) and organic carbon content analysis**. Total nitrogen (TN) and organic carbon (TOC)contents in decarbonated samples were measured at the Center of Modern Analysis, Nanjing University, using an Elementar Vario MICRO. Five bulk samples were selected for the total nitrogen ($TN_{bulk}$) measurements. N-contents compare well between bulk samples and the decarbonated residue (Supplementary Data 2), suggesting no significant loss of nitrogen during decalcification. The primary N isotopic composition preserved in the ancient sediments may have been modified during early and/or late burial diagenesis and metamorphism. The preservation of primary nitrogen isotopic signals could be assessed by examining the TOC, TN contents, TOC/N (atom) ratios, and their relationships with $\delta^{15}N$ data (see Supplementary Note 1 and Supplementary Fig. 2).

**Data availability**. The data that support the findings of this study are available from the corresponding authors upon reasonable request.

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

## Acknowledgements

This work was funded by the National Natural Science Foundation of China (41603023, 41661134048, 41430104), the CAGS Research Fund (YYWF201603), the Strategic Priority Research Program (B) of the Chinese Academy of Sciences (CAS) (XDB18000000), National Basic Research Program of China (2013CB835004), the Sino-German Geoscientific Research Project FOR 736, and the China Scholarship Council. G. S. acknowledges funding from NERC project NE/P013643/1. We thank Marianne Falk and Weiming Gong for their assistance with chemical analyses, and Prof. Junming Zhang, Dr. Michael Steiner, Guangyi Wei, Dr. Xi Chen, Dr. Da Li, Dr. Wei Wei for helpful discussions.

## Author contributions

D.W. and H.-F.L. designed the study, interpreted the data, and prepared the original manuscript. D.W., T.H., and M.Z. collected samples in the field. D.W., U.S., and A.G. carried out nitrogen and organic carbon isotopic analysis. G.A.S., U.S., and X.-K.Z. helped with interpreting the data and contributed to the writing of this manuscript. M.Z. and B.Y. contributed most of the stratigraphic correlations in the Supplementary Information. All authors discussed the results and commented on the manuscript.
