## [Peer Review File · Nature Communications]

Reviewers' comments:

Reviewer #1 (Remarks to the Author):

General comments to the authors:

The origin and diversification of animals in the middle-to-late Neoproterozoic and into the Paleozoic is frequently linked to contemporaneous changes in the oxygen content of the oceans and/or atmosphere. However, the problem of causation and its directionality (i.e. did animals drive these environmental changes as they diversified, or did these environmental changes allow animals to diversify in the first place?) remains difficult to resolve. In this manuscript, Wang and colleagues present N isotope data to reconstruct both changes in marine redox state and nutrient availability across the Ediacaran-Cambrian transition. The authors correlate these trends with major biological-evolutionary events. For example, the authors find that intervals characterized by relatively high $\delta^{15}\text{N}$ values -- interpreted to signal N cycling approximating today's -- correspond to major metazoan radiations, while intervals of relatively low $\delta^{15}\text{N}$ values correspond to extinctions, namely of the Ediacara biota and 'small shelly fossils.' These correlations are important to unraveling the causal connections between marine oxygen and nutrient availability, eukaryotic phytoplankton abundance, biological pump efficiency, and early animal diversification across the Neoproterozoic-Paleozoic transition. I find the authors' major interpretations to be reasonable, although a bit simplistic at times. In other words, while I find these geochemical data to be important, I don't think they resolve, by themselves, the central issue of causal directionality, even though the authors present a reasonable interpretation (enhanced N supply increased the energy flowing through eukaryotic ecosystems, encouraging the diversification of animals). Minor comments primarily concerning the background and discussion are detailed below.

Specific comments:

L 26-28: This statement is awkwardly worded.

L 44: Replace 'that reach' to 'approaching.'

L 45: "This" is an unclear pronoun. Are you referring to the abrupt appearance of soft-bodied multicellular organisms, or the replacement of these organisms by biomineralized animals? Delete 'figuratively.'

L 47-48: Rising oxygen isn't thought to have been a 'driver' per se, but rather the release of an environmental brake (i.e. the removal of a low-oxygen impasse). In other words, oxygen did not cause animals to diversify, oxygen allowed animals to diversify. Next, this geochemical data for oxygenation does not suggest that rising oxygen caused animal diversification -- all it says is that the two are correlated in time. This correlation, by itself, does not and cannot resolve the direction of causation, or whether the two were causally connected at all. Additional data (physiological, ecological, etc.) is needed resolve the issue of causation, as well as its directionality.

L 57: Delete 'biological.' Life is implied by 'metabolic.'

L 61: Replace 'prokaryotes' with 'bacteria and archaea.'

L 70: What is meant by 'competitiveness of eukaryotes'? The ability of algae to compete with cyanobacteria?

L 125-126: Add hyphen between 'small' and 'shelly,' and between 'Chegnjian' and 'type.'

L 132: Replace 'connection' with 'correlation.'

L 136: The oldest evidence for sponges, in the form of molecular biomarkers, dates back to the Sturtian-Marinoan interglacial period (Brocks et al., 2017 Nature), and recent molecular clock estimates place the origin of crown-group sponges in the Tonian (Dohrmann & Wörheide 2017 Scientific Reports). The oxygen requirements of sponges, therefore, are not obviously relevant or applicable to the Ediacaran and Cambrian diversification of animals, and, as presented, are set up as a sort of 'straw man.' That is, the oxygen requirements of sponges are more relevant to the origin of crown-group animals and crown-group sponges in the Tonian than to animal diversification in the Ediacaran and Cambrian. A better way to phrase this argument would be to mention that while the earliest animals (in the Tonian and Cryogenian) may have had relatively low oxygen requirements, Ediacaran and Cambrian levels of animal diversity likely required oxygen levels above those required to support the very first multicellular animals. For more details on these points, I refer the authors to Sperling, Knoll & Girguis 2015 Annu. Rev. Ecol. Evol. Syst., and Mills & Canfield 2014 Bioessays.

L 138: Emphasize the importance of ecological and genetic factors to what exactly?

L 159: What is meant by 'eumetazoans'? Sponges, which are not eumetazoans, are also predicted to have helped instigate this shift to larger eukaryotic phytoplankton (e.g. reference 38).

L 180: Replace 'animals in the higher level of the food chain' to 'animals occupying higher trophic levels.'

L 200: Delete 'mass.'

L 203: Typo – change to 'nutrient.' Also, what is meant by 'better'?

L 221: Insert 'oxygen' between 'modern' and 'levels.'

L 263: What is meant by 'evolutionary force'?

Reviewer #2 (Remarks to the Author):

Wang et al. present a new N-isotope dataset from the Yangtze Basin. The study is significant because this unique data set reveals significant coupling between $\delta^{15}\text{N}$ and $\delta^{13}\text{C}$ around the timing of evolutionary events occurring between 551 to 515 Ma. This new data set suggests that increase in N-fixation likely contributed to the metazoan radiations recorded in the fossil record at the time.

The authors make a valid point that increase in oceanic oxygen levels has been presumed to be the main driver behind the animal evolution. This view, however, might be simplistic, since there are still contradictory geochemical datasets that do not fully explain these events. Also there are still ongoing debates around the biology and oxygen requirements for early animals. Controversies around how the appearance of early animals, biological events, physiological needs and their link to geochemistry have not been fully settled. On the other hand, the topics regarding an increase in fixed nitrogen around this time, and how this might have facilitated episodic animal radiations reinforced by ocean oxygenation, have not received much attention - this study is therefore topical and should be published - I have no doubt the community will welcome this study.

The 'traditional' discussions have mainly emphasised oxygen, but seem to forget that nitrogen is a macronutrient. There is no doubt that increased N-fixation would have fuelled the biosphere. The authors need to make references to previous work proposing changes/links to the N and Carbon cycle in other fields (genomics/evolution and biomarkers). Such previous work complement the findings by Wang et al.

My main criticisms are: 1) Wang et al. have overlooked in their discussion some of the literature on biological events such as the origin of key planktonic groups and marine primary producers. The elements are there but it needs to be spelled out that there is some recent literature showing/suggesting a link between geochemistry (N, C), increase organic matter and the appearance of planktonic groups. I have made some suggestions below on literature that needs to be cited to complement their arguments. I suggest to add a section on marine primary producers (subheadings); 2) the discussion would benefit from having subheadings - at the moment it is a bit dense, and the arguments are hard to follow, 3) because Nature comms attracts a broad audience this manuscript needs a figure of the N-cycle with processes / redox - this will help follow the discussion.

Regarding methodologies is not my area of expertise, so the editor will need to check methodologies with an expert in this field of research. However, in my opinion obtaining isotopic data from shallow to deep marine environments seems the right approach to understand what was happening with the Nitrogen cycle at the time. Also the location of the samples is unique.

It is particularly exciting that episodes showing high $\delta^{15}\text{N}$ values are coincident with radiations of Ediacaran, and Chengjiang type biota. Also these episodes coincide with positive carbon isotope excursions! The timing of these events/fluctuations also coincide with the appearance or extinction of other fossils (biological events) suggesting that nutrient availability is tightly linked to the diversification of biota recorded in the fossil record. Wang et al. should also emphasise that different lines of evidence are converging into a similar explanation - this will make their argument stronger - this needs work.

In general the paper is well written. I recommend this manuscript to be published.

Specific comments:

167 - what does Ocean-going N-fixing cyanobacteria? This does not make any sense. Do you mean planktonic?

167-169 It is important to make clearer in this sentence that ancestors of planktonic nitrogen fixers radiated at this time (Sanchez-Baracaldo et al 2014) - While they use the correct reference, this sentence is confusing at the moment. The writing is similar to the commentary by Knoll 2017 - rewrite.

The discussion would benefit from adding recent evidence looking at the genomic/molecular clock analyses of planktonic groups - in addition to Sanchez-Baracaldo et al 2014, also add Sanchez-Baracaldo 2015 (Scientific Reports)¹ and Sanchez-Baracaldo et al 2017 PNAS². There is also evidence from biomarkers (Brocks, J. J. et al 2017³) suggesting that planktonic green algae, and ultimately a phototrophic eukaryotic component radiated prior to (551 to 515 Ma) events studied here. This is important because the radiation of planktonic groups (cyanobacteria, red (benthic) and green algae) would have contributed to increase marine primary productivity and increase if nitrogen fixation. In other words they would have generated the organic compounds needed to support life at this time. Here I would also discuss how the ideas of Butterfield and Lenton fit regarding primary producers.

The authors already make reference to the commentary by Knoll 2017 - but do not include Brocks, J. J. et al 2017- this needs to be added as it is the main paper Knoll is discussing in his commentary. Also some of the arguments are already been presented by Knoll - make sure citations are given accurately and fairly.

169-172 - This sentence needs rewriting - it is unclear what the authors mean by 'A large reservoir

of nitrogen, especially nitrate, was built up later in the Ediacaran - early Cambrian ocean due to enhanced N₂-fixation by N-fixing cyanobacteria under P-repleted conditions.'

172 --- Thus eukaryotic phytoplankton could have enjoyed a competitive advantage versus nitrogen-fixing prokaryotes.

What is the evidence? Or citation. Please look into preferential nitrogen sources for eukaryotes and bacteria- there is literature reporting eukaryotes preferentially using nitrates over ammonium. Also there are plenty of marine planktonic non-N-fixing cyanobacteria - evolving at this time. Eukaryotic and bacterial phytoplankton co-exist and there is a dynamic between communities possibly determined by nutrient availability. There is plenty of literature.

231-234 - Here the authors need to cite previous work by Sanchez-Baracaldo et al 2014 who already proposed that N availability would have promoted the biological pump at this time.

263-265 - Please rewrite: 'creating an evolutionary force to develop more complex living strategies and/or creating more space for further development of animal-based ecosystems.'

What do they mean by evolutionary force? Creating more space? I realise the authors are not biologists but these terminologies need to be applied properly - rather than adding them together because they sound good.

283. Figure 2. Could the authors explain what they mean by normal marine productivity (in green)?

1

Sánchez-Baracaldo, P. Origin of marine planktonic cyanobacteria. *Sci Rep-Uk* 5,, doi:10.1038/srep17418 (2015).

2 Sanchez-Baracaldo, P., Raven, J. A., Pisani, D. & Knoll, A. H. Early photosynthetic eukaryotes inhabited low-salinity habitats. *Proc Natl Acad Sci U S A* 114, E7737-E7745, doi:10.1073/pnas.1620089114 (2017).

3 Brocks, J. J. et al. The rise of algae in Cryogenian oceans and the emergence of animals. *Nature* 548, 578-581, doi:10.1038/nature23457 (2017).

Reviewer #3 (Remarks to the Author):

Review of "Coupling of ocean redox and animal evolution during the Ediacaran-Cambrian transition" by Wang et al.

The authors compile new and existing carbon and nitrogen isotope data from the Yangtze Basin across the Ediacaran-Cambrian boundary and infer consistent excursions in the two proxies across the basin that match important events in the fossil record of complex organisms. They conclude that a high supply of fixed nitrogen and phosphorus – coupled through feedback mechanisms – may have facilitated pulses in animal diversification at this time.

Extensive research has already been carried out on the carbon and nitrogen cycle in the Yangtze

Basin. The novelty of this manuscript is partly the contribution of a new data set from another site, and more importantly, the recognition that basin-wide excursions in the two isotopic proxies coincide with events in animal evolution. This topic is currently of wide interest, and I would therefore expect this paper to be widely cited.

The new data appear to be of good quality (although the inclusion of a standard for accuracy would be important), and possible alteration effects of diagenesis and metamorphism are adequately addressed in the supplements. The more detailed data analysis would be strengthened with statistical tests to show that the inferred isotopic excursions are indeed significant.

A major concern is with the proposed 'see-saw' mechanism of anoxic and oxic events (ll. 191-221). Several points need to be addressed:

(a) The authors state that under both anoxic and oxic conditions nitrogen and phosphorus get trapped in sediments and become limiting (ll. 195-196 and ll. 212-213). Would this imply that the two nutrients can only accumulate under transient intermediate conditions? This is unclear.

(b) The chicken-and-egg problem is not fully resolved, because the termination of anoxic events is explained by increasing bioturbation (l. 211), which would imply a biological trigger. However, how were these bioturbating organisms able to expand while the ocean was still anoxic? A few more sentences would help here.

(c) The text mentions 'short-term' feedback loops; however, the oxic and anoxic intervals alternate on timescales of several million years as shown in Figure 2. Is it realistic that the build-up of reductants or oxidants and the burial of nutrients would take that long? Thousand-year timescales, corresponding to ocean-mixing timescales, would seem more likely for this proposed mechanism. Perhaps consider external triggers (see next comment).

(d) The transition from alternating oxic-anoxic events to a fully oxic ocean (l. 220) is not well explained, because it is not clear why the proposed mechanism would lead to a progression towards more oxic conditions. Is the idea that with the appearance of macro-fauna it would become impossible to go back into an anoxic state? But if so, how would that explain anoxic events in the Phanerozoic? I realize that this discussion may go beyond the scope of the manuscript. Perhaps one possible solution to this and some of the previous points would be an external driver, such as climatic perturbations. See, for example, Zhai et al. 2018 (Paleo3), who proposed global cooling and warming events across the Ediacaran-Cambrian transition as a driver of ocean circulation and productivity.

Detailed comments:

l. 61: change to 'performed by some prokaryotes'. Otherwise it may give the impression that all prokaryotes fix nitrogen.

ll. 133-157: This proposed linkage between iron speciation and nitrogen isotopes is provocative and an interesting idea. Another possible explanation for the seemingly 'anoxic' iron signature may be that Fe²⁺ upwelled from the anoxic abyssal deep ocean into the Yangtze Basin and got trapped there as authigenic iron as it encountered oxic conditions. This scenario is exemplified by the Peruvian upwelling zone, where authigenic iron is actually slightly depleted within the OMZ but enriched in the oxic sediments immediately underneath (Scholz et al. 2014 GCA). To test this hypothesis it may be worth checking the exact speciation of iron in these sediments.

ll. 164-166: Note that P bioavailability also depends on the supply of oxidants for remineralization of organic matter (Kipp & Stüeken 2017 Science Advances). It is not certain that enhanced P delivery by weathering could on its own act as a trigger for enhanced N₂ fixation. I would suggest rephrasing this part of the text to account for these uncertainties.

I. 169: The build-up of this nitrate reservoir would also have required the establishment of oxic conditions. That should be stressed here.

I. 171-172: It's not clear why eukaryotic phytoplankton would have had a competitive advantage over N₂-fixing prokaryotes – they would still have depended on them. It should be sufficient to say that eukaryotic phytoplankton were able to thrive under these conditions. Perhaps cite Brocks et al. 2017 (Nature), who provide biomarker evidence for the radiation of eukaryotic algae at this time.

I. 194: Delete 'fortunately'. That's a bit of a subjective term.

II. 201-221: The writing style is noticeably different in this part of the manuscript. I suggest rephrasing some of the sentences for clarity.

I. 214: It is true that a nitrogen cycle dominated by sedimentary denitrification would show d¹⁵N values around 0 permil (Quan & Falkowski 2009 Geobiology, Figure 4). However, in an ocean that has oxygen minimum zones somewhere, this is an unlikely scenario. Even the modern ocean shows the effects of water-column denitrification in locally suboxic regions. These light d¹⁵N values are more parsimoniously interpreted as an N₂-fixation dominated nitrogen cycle.

I. 247-257: This section on correlative data from Kazakhstan is a strength of the paper. I would suggest moving it further up, closer to I. 222 where the global significance of the data is first discussed.

I. 258-259: The text mentions a trend here in d¹⁵N that indicates progressive oxygenation into the Cambrian. However, in Figure 2, the Cambrian d¹⁵N values are actually similar or even lighter than d¹⁵N values in the Ediacaran. This needs to be reconciled with the inferred oxygenation. I'm wondering if some of the light d¹⁵N data in the Cambrian sections reflect local paleogeography (restricted basins). Is that possible?

Lastly, the authors mention phosphate deposition in the basin (II. 84-85). Are there temporal or spatial trends in phosphate abundances that could be integrated into the discussion?

I hope, these comments will be helpful with the publication of the manuscript.

Best regards,
Eva Stüeken

The line numbers mentioned below all refer to the marked manuscript.

Replies to the Reviewer #1

Reviewer #1 (Remarks to the Author):

General comments to the authors:

Comments #1: The origin and diversification of animals in the middle-to-late Neoproterozoic and into the Paleozoic is frequently linked to contemporaneous changes in the oxygen content of the oceans and/or atmosphere. However, the problem of causation and its directionality (i.e. did animals drive these environmental changes as they diversified, or did these environmental changes allow animals to diversify in the first place?) remains difficult to resolve. In this manuscript, Wang and colleagues present N isotope data to reconstruct both changes in marine redox state and nutrient availability across the Ediacaran-Cambrian transition. The authors correlate these trends with major biological-evolutionary events. For example, the authors find that intervals characterized by relatively high $\delta^{15}\text{N}$ values -- interpreted to signal N cycling approximating today's -- correspond to major metazoan radiations, while intervals of relatively low $\delta^{15}\text{N}$ values correspond to extinctions, namely of the Ediacara biota and 'small shelly fossils.' These correlations are important to unraveling the causal connections between marine oxygen and nutrient availability, eukaryotic phytoplankton abundance, biological pump efficiency, and early animal diversification across the Neoproterozoic-Paleozoic transition. I find the authors' major interpretations to be reasonable, although a bit simplistic at times. In other words, while I find these geochemical data to be important, I don't think they resolve, by themselves, the central issue of causal directionality, even though the authors present a reasonable interpretation (enhanced N supply increased the energy flowing through eukaryotic ecosystems, encouraging the diversification of animals). Minor comments primarily concerning the background and discussion are detailed below.

Revisions/Response: Many thanks for all these constructive comments.

The conventional view is that oxygenation, and in particular the oxygenation of the oceans, was the main trigger behind the evolutionary radiation of animals during the Ediacaran-Cambrian transition (Knoll, 2003; Nursall, 1959). However, it is increasingly argued that some of the most primitive animals (such as sponges) could have survived in a low oxygen environment and may themselves have played a key role in regulating oxygen levels since at least the Ediacaran Period (Lenton et al., 2014; Mills et al., 2014). Resolving this chicken-and-egg issue remains difficult but doing so lies outside the scope of our study.

In our study, we identify a correlation between nitrate availability, ocean oxygenation and animal bioradiations in the late Ediacaran-early Cambrian interval. We cautiously draw the conclusion that pulses of deep water oxygenation removed physiological barriers to - or facilitated - biological innovations among Ediacaran- and early Cambrian- type animals, rather than being a direct driver of

the Ediacaran- and early Cambrian- bioradiations (lines 177-180 and 217-219). Furthermore, in spite of the importance of oxygen, we also emphasize that increased nutrient supply, especially nitrogen, may have exerted a major control on both ocean oxygenation and macroevolution, by allowing large-celled, eukaryotic phytoplankton to flourish and so create a more efficient biological pump (line 190-239).

Specific comments:

Comments #2: L 26-28: This statement is awkwardly worded.

Revisions/Response: This sentence has been reworded / revised to “The role of the physical environment in this biological revolution, such as changes to oxygen levels and nutrient availability, has been the focus of longstanding debate. Seemingly contradictory data from geochemical redox proxies help to fuel this controversy” (line 26-30).

Comments #3: L 44: Replace ‘that reach’ to ‘approaching.’

Revisions/Response: Revised as suggested.

Comments #4: L 45: “This” is an unclear pronoun. Are you referring to the abrupt appearance of soft-bodied multicellular organisms, or the replacement of these organisms by biomineralized animals? Delete ‘figuratively.’

Revisions/Response: Revised as suggested.

Here the word “this” refers to both the radiations of soft-bodied multicellular organisms in the late Ediacaran and biomineralized animals in the early Cambrian. To make it clear, the sentence has been revised to “The underlying causes for biodiversification and bioradiations during the Ediacaran and early Cambrian are still controversial” (line 47-49).

Comments #5: L 47-48: Rising oxygen isn’t thought to have been a ‘driver’ per se, but rather the release of an environmental brake (i.e. the removal of a low-oxygen impasse). In other words, oxygen did not cause animals to diversify, oxygen allowed animals to diversify. Next, this geochemical data for oxygenation does not suggest that rising oxygen caused animal diversification – all it says is that the two are correlated in time. This correlation, by itself, does not and cannot resolve the direction of causation, or whether the two were causally connected at all. Additional data (physiological, ecological, etc.) is needed resolve the issue of causation, as well as its directionality.

Revisions/Response: It is indeed difficult to determine the causal direction between oxygenation and early animal diversification. Thus, the use of word ‘driver’ is indeed inappropriate here. In the discussion section of this manuscript, we cautiously draw the conclusion that pulses in deep water oxygenation removed physiological barriers to, or facilitated, biological innovations among benthic Ediacaran- and Cambrian- type animals, rather than triggering the Ediacaran- and early Cambrian- bioradiations. Furthermore, we emphasize the importance of

nitrogen supply to both ocean oxygenation and macroevolution (Please refer to our reply to **Comments #1** of **Reviewer #1**).

Accordingly, the original sentence here has been revised into “The key innovations in complex animals during the Ediacaran–Cambrian transition interval were skeletonization⁴, motility⁵ and predation⁶, large body-size and complex food webs^{1,2}. While some simple animals, such as soft-bodied sponges, which may have their origins in the Tonian⁷, could have had low oxygen requirements (0.5–4.0% of present-day levels)⁸, macroscopic animals of the Ediacaran–early Cambrian transition likely required oxygen levels above the physiological thresholds for those energetically expensive behaviors^{9,10}” (line 49-56).

Furthermore, in the later sections of this manuscript, in order to discuss the importance of nitrogen supply to both ocean oxygenation and macroevolution, we added some other evidence, such as molecular biomarkers on the evolution of primary producers, molecular clock estimates of the origin of planktonic algae and earliest animals (sponges), and ecological factors (e.g. predation) in the revised manuscript (line 190-239).

Comments #6: L 57: Delete ‘biological.’ Life is implied by ‘metabolic.’
Revisions/Response: Revised as suggested.

Comments #7: L 61: Replace ‘prokaryotes’ with ‘bacteria and archaea.’
Revisions/Response: Revised as suggested.

Comments #8: L 70: What is meant by ‘competitiveness of eukaryotes’? The ability of algae to compete with cyanobacteria?
Revisions/Response: The original meaning of this phrase is indeed to emphasize the ability of algae to compete with cyanobacteria under nutrient-replete conditions. However, as is pointed out by **Comment #12** of **Reviewer #2**, eukaryotic and bacterial phytoplankton could co-exist in the ocean and their distributions are influenced by nutrient availability (Sanchez-Baracaldo, 2015, Flombaum et al., 2013). Bacterial phytoplankton either have the ability for N₂-fixation or preferential ammonia assimilation. However, eukaryotic phytoplankton cannot fix nitrogen and preferentially assimilate nitrate over ammonia (Fawcett et al., 2011). Thus, eukaryotic phytoplankton tend to flourish where nutrient concentrations are high. The explanation of this point has been added in the revised manuscript (line 206-211).

Here, the original sentence has been revised to: “As it is a principal nutrient for life, nitrogen, by way of its redox sensitivity is considered to be an important control on the growth of eukaryotic producers and primary productivity” (line 80-83)

Comments #9: L 125-126: Add hyphen between ‘small’ and ‘shelly,’ and between

'Chenggjiang' and 'type.'

Revisions/Response: Revised as suggested.

Comments #10: L 132: Replace 'connection' with 'correlation.'

Revisions/Response: Revised as suggested.

Comments #11: L 136: The oldest evidence for sponges, in the form of molecular biomarkers, dates back to the Sturtian-Marinoan interglacial period (Brocks et al., 2017 Nature), and recent molecular clock estimates place the origin of crown-group sponges in the Tonian (Dohrmann & Wörheide 2017 Scientific Reports). The oxygen requirements of sponges, therefore, are not obviously relevant or applicable to the Ediacaran and Cambrian diversification of animals, and, as presented, are set up as a sort of 'straw man.' That is, the oxygen requirements of sponges are more relevant to the origin of crown-group animals and crown-group sponges in the Tonian than to animal diversification in the Ediacaran and Cambrian. A better way to phrase this argument would be to mention that while the earliest animals (in the Tonian and Cryogenian) may have had relatively low oxygen requirements, Ediacaran and Cambrian levels of animal diversity likely required oxygen levels above those required to support the very first multicellular animals. For more details on these points, I refer the authors to Sperling, Knoll & Girguis 2015 Annu. Rev. Ecol. Evol. Syst., and Mills & Canfield 2014 Bioessays.

Revisions/Response: Revised as suggested.

As is suggested by this comment, this sentence is not quite relevant to oxygenation and animal diversification. Thus, the original sentence has been deleted in the revised manuscript. In the meantime, a description of oxygen requirements of sponges and Ediacaran- and Cambrian- type animals has been added in the beginning of the revised manuscript (line 52-56, please refer to our reply to **Comment #1 and #5 of Reviewer #1**).

The recommended references, Brocks et al., 2017, Dohrmann & Wörheide 2017, Sperling et al., (2015) and Mills & Canfield 2014 has been cited in the revised manuscript (e.g., line 47-49, 52-56, 201-205 and 236-238).

Comments #12: L 138: Emphasize the importance of ecological and genetic factors to what exactly?

Revisions/Response: As this is not quite relevant to the topic of this paragraph, the original sentence has been deleted in the revised manuscript (please refer to our reply to **Comment #11 of Reviewer #1**).

In the meantime, we have added some discussion on the importance of ecological and genetic factors to animal evolution and ocean oxygenation in the revised manuscript (line 230-239).

"Filter-feeding sponges evolved earlier, perhaps already in the Tonian Period when

oxygen levels in seawater had not yet significantly increased⁷ (**genetic factors**). Sponges probably fed on free organic matter and picoplankton, which did place a selective pressure for large-sized eukaryotic phytoplankton⁴⁷ (**ecological factors**). However, top-down ecological pressure may not have been enough because the bloom of large eukaryotic phytoplankton still require a favourable environment with sufficient nutrient availability (e.g., nitrate and phosphorus) as a prerequisite. Thus, even though it may be difficult to determine the causal directionality between ocean oxygenation and origin of animals³, our data demonstrate that marine nutrient availability probably exerted an important control on both ocean oxygenation and macroevolution.”

Comments #13: L 159: What is meant by ‘eumetazoans’? Sponges, which are not eumetazoans, are also predicted to have helped instigate this shift to larger eukaryotic phytoplankton (e.g. reference 38).

Revisions/Response: The ability of sponges to filter feed has been considered and added in the revised manuscript. The sentence has therefore been revised to “The feeding pressure of animals, such as filtering and grazing, was suggested to contribute to the evolution of large-celled, eukaryotic phytoplankton as an important force” (line 223-225).

Comments #14: L 180: Replace ‘animals in the higher level of the food chain’ to ‘animals occupying higher trophic levels.’

Revisions/Response: Revised as suggested.

Comments #15: L 200: Delete ‘mass.’

Revisions/Response: Revised as suggested.

Comments #16: L 203: Typo – change to ‘nutrient.’ Also, what is meant by ‘better’?

Revisions/Response: The word “nitrients” has been corrected, and “better” has been replaced by “efficiently”.

Comments #17: L 221: Insert ‘oxygen’ between ‘modern’ and ‘levels.’

Revisions/Response: Revised as suggested.

Comments #18: L 263: What is meant by ‘evolutionary force’?

Revisions/Response: The original sentence here has been revised to “These redox fluctuations were probably of global significance and closely associated with changes in nutrient abundances on geological time scales. We further propose that increased nutrient (such as nitrogen) availability could have exerted an important control on both oxygenation and macroevolution during the late Ediacaran–early Cambrian transition, by boosting the primary productivity of large-celled, eukaryotic phytoplankton in the ocean. The sporadic ocean anoxic events along continental margins may have been closely related to extinctions of Ediacaran- and some early Cambrian- type animals” (line 385-395).

Replies to the Reviewer #2

Reviewer #2 (Remarks to the Author):

Comments #1: Wang et al. present a new N-isotope dataset from the Yangtze Basin. The study is significant because this unique data set reveals significant coupling between $\delta^{15}\text{N}$ and $\delta^{13}\text{C}$ around the timing of evolutionary events occurring between 551 to 515 Ma. This new data set suggests that increase in N-fixation likely contributed to the metazoan radiations recorded in the fossil record at the time.

The authors make a valid point that increase in oceanic oxygen levels has been presumed to be the main driver behind the animal evolution. This view, however, might be simplistic, since there are still contradictory geochemical datasets that do not fully explain these events. Also there are still ongoing debates around the biology and oxygen requirements for early animals. Controversies around how the appearance of early animals, biological events, physiological needs and their link to geochemistry have not been fully settled. On the other hand, the topics regarding an increase in fixed nitrogen around this time, and how this might have facilitated episodic animal radiations reinforced by ocean oxygenation, have not received much attention - this study is therefore topical and should be published - I have no doubt the community will welcome this study.

Revisions/Response: Many thanks for the constructive comments / criticisms.

Based on the current knowledge of genomics, ecology and geochemical datasets, it is indeed difficult to determine the causal direction between ocean oxygenation and early animal diversification. Therefore, in the revised manuscript, we have emphasized that episodic ocean oxygenation during the Ediacaran–early Cambrian interval removed physiological barriers to biological innovations among benthic Ediacaran- and Cambrian- type animals, rather than being a direct driver of the bio-radiations (lines 177-180 and 217-219). The most important contribution of our study is that we reveal convincing correlation between nitrate availability and bio-radiations. We propose that increased nitrogen supply (especially nitrate) exerted an important control on both ocean oxygenation and macroevolution, facilitating the flourishing of large-celled, eukaryotic phytoplankton to create a more efficient biological pump (line 190-239). Please also see our reply to **Comment #1 of Reviewer #1**.

Comments #2: The 'traditional' discussions have mainly emphasised oxygen, but seem to forget that nitrogen is a macronutrient. There is no doubt that increased N-fixation would have fuelled the biosphere. The authors need to make references to previous work proposing changes/links to the N and Carbon cycle in other fields (genomics/evolution and biomarkers). Such previous work complement the findings by Wang et al.

Revisions/Response: Revised as suggested. Some valuable references on biomarkers and phylogenomics have been added in the revised manuscript, including Brocks et al., 2017, Sánchez-Baracaldo, 2015 and Sánchez-Baracaldo et al., 2017. Please refer to our reply to **Comment #3 of Reviewer #2** for detailed

explanations.

My main criticisms are:

Comments #3: 1) Wang et al. have overlooked in their discussion some of the literature on biological events such as the origin of key planktonic groups and marine primary producers. The elements are there but it needs to be spelled out that there is some recent literature showing/suggesting a link between geochemistry (N, C), increase organic matter and the appearance of planktonic groups. I have made some suggestions below on literature that needs to be cited to complement their arguments. I suggest to add a section on marine primary producers (subheadings);

Revisions/Response: We have now added a detailed discussion on the evolution of marine primary producers in a dedicated paragraph in the Section “Ocean oxygenation and biological evolution”, as well as valuable references (line 190-222).

Firstly, planktonic N-fixing cyanobacteria probably radiated into the open ocean during the late Neoproterozoic to make a major contribution to fixed nitrogen supply in seawater (Sánchez-Baracaldo et al., 2014; Sánchez-Baracaldo, 2015). The increase in nutrient availability modified the ecological structure of primary producers, likely facilitating a shift in phytoplankton composition from small-celled, cyanobacterial picoplankton towards large-celled, planktonic algae in the Neoproterozoic (Brocks et al., 2017; Sánchez-Baracaldo et al., 2017). Finally, large-cell eukaryotic phytoplankton were able to thrive under nitrate-replete conditions in the Ediacaran–early Cambrian interval, as diffusion limitation of nutrient uptake increases with cell size. This evolution of marine primary producers is closely linked with biogeochemical nitrogen and carbon cycles, and has created a more efficient biological pump, thus making a major contribution to both ocean oxygenation and animal innovations (see line 190-222).

Comments #4: 2) the discussion would benefit from having subheadings - at the moment it is a bit dense, and the arguments are hard to follow,

Revisions/Response: Revised as suggested.

We have added four subheadings in the revised manuscript, including “Ocean oxygenation and biological evolution” (line 152), “Dynamic redox fluctuations and nutrient regulation” (line 241), “Global significance” (line 324) and “Summary” (line 380).

Comments #5: 3) because Nature comms attracts a broad audience this manuscript needs a figure of the N-cycle with processes / redox - this will help follow the discussion.

Revisions/Response: Revised as suggested. A schematic diagram for the biogeochemical nitrogen cycle in the late Ediacaran–early Cambrian has been added in the revised manuscript, and named as Figure 3 (line 417-423)

Comments #6: Regarding methodologies is not my area of expertise, so the editor will need to check methodologies with an expert in this field of research. However, in my opinion obtaining isotopic data from shallow to deep marine environments seems the right approach to understand what was happening with the Nitrogen cycle at the time. Also the location of the samples is unique.

It is particularly exciting that episodes showing high $\delta^{15}\text{N}$ values are coincident with radiations of Ediacaran, and Chengjiang type biota. Also these episodes coincide with positive carbon isotope excursions! The timing of these events/fluctuations also coincide with the appearance or extinction of other fossils (biological events) suggesting that nutrient availability is tightly linked to the diversification of biota recorded in the fossil record. Wang et al. should also emphasise that different lines of evidence are converging into a similar explanation - this will make their argument stronger - this needs work.

Revisions/Response: Revised as suggested. Different lines of evidence, such as biomarker data, fossil records, carbon and nitrogen isotope data, have been combined together, to emphasize our proposal that increased nutrient supply has probably exerted a major control on both ocean oxygenation and macroevolution.

The current biomarker and molecular clock data suggested that planktonic N-fixing cyanobacteria radiated into the open ocean during the late Neoproterozoic (Sanchez-Baracaldo, 2015, Sanchez-Baracaldo et al, 2014), followed by the rise of eukaryotic phytoplankton (Brocks et al., 2017; Sanchez-Baracaldo et al, 2017). This is consistent with our high $\delta^{15}\text{N}$ values in intervals I, III and V, which suggest that a stable nitrate reservoir existed widely in the substantially oxygenated ocean in the late Ediacaran-early Cambrian, facilitating the bloom of large-celled eukaryotic phytoplankton. Also, the radiation of eukaryotic phytoplankton created a more efficient biological pump, which is consistent with positive carbon isotope values in intervals I, III and V. Therefore, increased nitrogen supply by planktonic N-fixing cyanobacteria facilitated the rise of eukaryotic phytoplankton, leading to pulses of oxygenation in deep waters during intervals I, III and V, which removed physiological barriers for bioradiations of benthic Ediacaran- and Cambrian- type animals. Therefore, we could draw the conclusion that increased nutrient supply probably exerted a major control on both ocean oxygenation and macroevolution.

This detailed explanation has been added in the Section "Ocean oxygenation and biological evolution", lines 190-222 of the revised manuscript.

In general the paper is well written. I recommend this manuscript to be published.
Thanks!

Specific comments:

Comments #6: 167 - what does Ocean-going N-fixing cyanobacteria? This does make any sense. Do you mean planktonic?

Revisions/Response: Yes, the word “ocean-going” does mean “planktonic” and has been revised to “planktonic” in the revised manuscript (line 190).

Comments #8: 167-169 It is important to make clearer in this sentence that ancestors of planktonic nitrogen fixers radiated at this time (Sanchez-Baracaldo et al 2014) - While they use the correct reference, this sentence is confusing at the moment. The writing is similar to the commentary by Knoll 2017 - rewrite.

Revisions/Response: Revised as suggested.

This sentence has been revised to “As major diazotrophs, planktonic N-fixing cyanobacteria have been suggested to radiate into the open ocean during the late Neoproterozoic, substantially changing nitrogen supply in seawater^{36, 37} in response to a putative rise in phosphorus availability³⁸” (line 190-195). The valuable reference Sanchez-Baracaldo (2015) has also been added here.

As the discussion on the evolution of primary producers is fairly important (please see our reply to **Comment #3 of Reviewer #2**), this sentence has been moved up to the beginning of the second paragraph in the Section “Ocean oxygenation and biological evolution”.

Comments #9: The discussion would benefit from adding recent evidence looking at the genomic/molecular clock analyses of planktonic groups - in addition to Sanchez-Baracaldo et al 2014, also add Sanchez-Baracaldo 2015 (Scientific Reports)¹ and Sanchez-Baracaldo et al 2017 PNAS². There is also evidence from biomarkers (Brocks, J. J. et al 2017³) suggesting that planktonic green algae, and ultimately a phototrophic eukaryotic component radiated prior to (551 to 515 Ma) events studied here. This is important because the radiation of planktonic groups (cyanobacteria, red (benthic) and green algae) would have contributed to increase marine primary productivity and increase if nitrogen fixation. In other words they would have generated the organic compounds needed to support life at this time. Here I would also discuss how the ideas of Butterfield and Lenton fit regarding primary producers.

Revisions/Response: Revised as suggested.

Firstly, we have now added detailed discussion on the evolution of marine primary producers in a dedicated paragraph at the Section “Ocean oxygenation and biological evolution” and cited the valuable references Sanchez-Baracaldo, 2015, Sanchez-Baracaldo et al, 2017 and Brocks et al., 2017 (line 190-222). Please refer to our reply to **Comment #3 of Reviewer #2**

Secondly, we have added the discussion on how the ideas of Butterfield and Lenton fit regarding primary producers. Butterfield et al. (2009) proposed that suspension-feeding animals contributed to the evolution of large-celled eukaryotic phytoplankton. However, the earliest suspension-feeding animals are reported not to appear until the early Cambrian (510–515 Ma, Harvey and Butterfield, 2008). Lenton et al. (2014) have suggested that filter-feeding sponges

probably fed on the picoplankton and dissolved organic carbon, thus providing a selective pressure for the evolution of large-sized eukaryotic phytoplankton. This scenario may be possible, because sponges did evolve earlier, perhaps already in the Tonian period when oxygen levels in seawater were lower (Dohrmann and Wörheide, 2017). However, top-down ecological pressure may be not enough because bloom of large eukaryotic phytoplankton still require a favourable environment with efficient nutrient availability (such as nitrate and phosphorus) as a prerequisite. Thus we propose that marine nutrient availability exerted a major control on both ocean oxygenation and macroevolution. The detailed explanation has been added in the Section “Ocean oxygenation and biological evolution” (line 223-239).

Comments #10: The authors already make reference to the commentary by Knoll 2017 - but do don't include Brocks, J. J. et al 2017- this needs to be added as it is the main paper Knoll is discussing in his commentary. Also some of the arguments are already been presented by Knoll - make sure citations are given accurately and fairly.

Revisions/Response: Revised as suggested.

The references Brocks. et al 2017 and Knoll et al., 2017 have been correctly cited in the second paragraph of the Section “Ocean oxygenation and biological evolution” in the revised manuscript.

e.g.

“As major diazotrophs, planktonic N-fixing cyanobacteria have been suggested to radiate into the open ocean during the late Neoproterozoic, substantially changing nitrogen supply in seawater (Sanchez-Baracaldo, 2015, Sanchez-Baracaldo et al, 2014), in response to a putative rise in phosphorus availability (Knoll et al., 2017)” (line 190-195)

“Enhanced supply of these bio-limiting nutrients (phosphorus and nitrogen) therefore modified the ecological structure of primary producers, likely facilitating a shift in phytoplankton composition from small-celled, cyanobacterial picoplankton towards large-celled, planktonic algae in the Neoproterozoic as evidenced from biomarker and molecular clock data (Brocks et al., 2017; Sanchez-Baracaldo et al, 2017)” (line 201-205)

Comments #11: 169-172 - This sentence needs rewriting - it is unclear what the authors mean by 'A large reservoir of nitrogen, especially nitrate, was built up later in the Ediacaran - early Cambrian ocean due to enhanced N₂-fixation by N-fixing cyanobacteria under P-repleted conditions.

Revisions/Response: Revised as suggested.

This sentence has been revised to “a stable reservoir of nitrate widely existed in the substantially oxygenated ocean of the late Ediacaran–early Cambrian, as revealed by our $\delta^{15}\text{N}$ data for the time intervals I, III and V” (line 206-208).

Comments #12: 172 - Thus eukaryotic phytoplankton could have enjoyed a competitive advantage versus nitrogen-fixing prokaryotes.

What is the evidence? Or citation. Please look into preferential nitrogen sources for eukaryotes and bacteria- there is literature reporting eukaryotes preferentially using nitrates over ammonium. Also there are plenty of marine planktonic non-N-fixing cyanobacteria - evolving at this time. Eukaryotic and bacterial phytoplankton co-exist and there is a dynamic between communities possibly determined by nutrient availability. There is plenty of literature.

Revisions/Response: Indeed, eukaryotic and bacterial phytoplankton could co-exist in the oceans and their distributions are influenced by nutrient availability (Sanchez-Baracaldo, 2015, Flombaum et al., 2013). Bacterial phytoplankton can undergo N₂-fixation or preferentially assimilate ammonia, while eukaryotic phytoplankton cannot fix nitrogen and preferentially assimilate nitrate over ammonia (Fawcett et al., 2011). Thus, bacterial phytoplankton dominate under oligotrophic conditions. On the contrary, larger eukaryotic phytoplankton tend to flourish where nutrient concentrations are high. Hence the original sentence here has been revised to “Large-celled, eukaryotic phytoplankton preferentially assimilate nitrate over ammonia⁴⁴, and so were able to thrive under nitrate-replete conditions, as diffusion limitation of nutrient uptake increases with cell size⁴⁵” (line 208-211).

Comments #13: 231-234 - Here the authors need to cite previous work by Sanchez-Baracaldo et al 2014 who already proposed that N availability would have promoted the biological pump at this time.

Revisions/Response: Revised as suggested. The reference Sanchez-Baracaldo et al 2014 has been added here.

Comments #14: 263-265 - Please rewrite: 'creating an evolutionary force to develop more complex living strategies and/or creating more space for further development of animal-based ecosystems.'

What do they mean by evolutionary force? Creating more space? I realise the authors are not biologists but these terminologies need to be applied properly - rather than adding them together because they sound good.

Revisions/Response: Revised as suggested.

The original sentence here has been revised to “These redox fluctuations were probably of global significance and closely associated with changes in nutrient abundances on geological time scales. We further propose that increased nutrient (such as nitrogen) availability could have exerted an important control on both oxygenation and macroevolution during the late Ediacaran–early Cambrian transition, by boosting the primary productivity of large-celled, eukaryotic phytoplankton in the ocean. The sporadic ocean anoxic events along continental margins may have been closely related to extinctions of Ediacaran- and some early Cambrian- type animals” (Section “Summary”, line 385-395)

Comments #15: 283. Figure 2. Could the authors explain what they mean by normal marine productivity (in green)?

Revisions/Response: The “normal marine production” means that marine production by primary producers is based on nitrate as the dominant nutrient, as in modern ocean, with $\delta^{15}\text{N}$ values ranging from +2‰ to +8‰. To make it more clear, this phrase has been revised to “Nitrate-based production as in modern ocean”. The explanation has been added in the caption of Fig.2 (line 413-415).

1 Sánchez-Baracaldo, P. Origin of marine planktonic cyanobacteria. *Sci Rep-Uk* 5,, doi:10.1038/srep17418 (2015).

2 Sanchez-Baracaldo, P., Raven, J. A., Pisani, D. & Knoll, A. H. Early photosynthetic eukaryotes inhabited low-salinity habitats. *Proc Natl Acad Sci U S A* 114, E7737-E7745, doi:10.1073/pnas.1620089114 (2017).

3 Brocks, J. J. et al. The rise of algae in Cryogenian oceans and the emergence of animals. *Nature* 548, 578-581, doi:10.1038/nature23457 (2017).

Replies to the Reviewer #3

Reviewer #3 (Remarks to the Author):

Review of “Coupling of ocean redox and animal evolution during the Ediacaran-Cambrian transition” by Wang et al.

The authors compile new and existing carbon and nitrogen isotope data from the Yangtze Basin across the Ediacaran-Cambrian boundary and infer consistent excursions in the two proxies across the basin that match important events in the fossil record of complex organisms. They conclude that a high supply of fixed nitrogen and phosphorus – coupled through feedback mechanisms – may have facilitated pulses in animal diversification at this time.

Extensive research has already been carried out on the carbon and nitrogen cycle in the Yangtze Basin. The novelty of this manuscript is partly the contribution of a new data set from another site, and more importantly, the recognition that basin-wide excursions in the two isotopic proxies coincide with events in animal evolution. This topic is currently of wide interest, and I would therefore expect this paper to be widely cited.

Many thanks for the constructive comments

Comments #1: The new data appear to be of good quality (although the inclusion of a standard for accuracy would be important), and possible alteration effects of diagenesis and metamorphism are adequately addressed in the supplements. The more detailed data analysis would be strengthened with statistical tests to show that the inferred isotopic excursions are indeed significant.

Revisions/Response: Revised as suggested.

We measured the $\delta^{15}\text{N}$ compositions in the laboratories of the “Museum für Naturkunde” in Berlin, using a Peptone house-standard whose value is $\delta^{15}\text{N} = +7.6\text{‰}$. The typical analytical precision for peptone is better than 0.15‰ (1 standard deviation) during the analyses of samples of this study. However, the typical analytical precision for sediments could be worse regarding the repeated analyses. For that purpose, the so-called “round robin test” has been conducted in many laboratories, analyzing sediment samples with unknown isotope compositions and very different nitrogen contents (Bahlmann et al., 2010). In this test, the laboratories of the “Museum für Naturkunde” were included and it turned out that the overall precision for sediments was a bit worse than that for peptone (0.15‰), but still better than the 0.3‰ (1 standard deviation). The overall variations of the nitrogen isotope records in our study are approximately 10 to 15 times larger than the error published in the Round Robin test. It means that our data are reproducible and of high quality. The detailed explanation of this point has been added into the “Method” in the revised manuscript (line 433-436).

In order to statistically determine whether the inferred nitrogen isotopic excursions are indeed significant, a student t-test was performed in five groups (I–

II, II-III, III-IV, IV-V and V-VI).

Our null hypothesis is:

No differences between positive and negative $\delta^{15}\text{N}$ excursions

When p-value is lower than 0.05, the null hypothesis can be rejected.

The results show that p-values are much lower than 0.05 (see the figure below). In other words, we can be reasonably confident that differences between positive and negative $\delta^{15}\text{N}$ values are highly significance. This conclusion has been added to the caption of Fig. 2 in the revised manuscript (line 412-413).

Comments #2: A major concern is with the proposed ‘see-saw’ mechanism of anoxic and oxic events (ll. 191-221). Several points need to be addressed:

(a) The authors state that under both anoxic and oxic conditions nitrogen and phosphorus get trapped in sediments and become limiting (ll. 195-196 and ll. 212-213). Would this imply that the two nutrients can only accumulate under transient intermediate conditions? This is unclear.

Revisions/Response:

Under oxic conditions, phosphorus (phosphate) will be co-precipitated with iron (hydr)oxides or adsorbed onto their surfaces, and hence preferentially retained in the sediments (Filippelli, 2008). Under anoxic conditions, phosphorus (1) is preferentially lost relative to organic carbon from organic matter (Ingall et al.,

2005); and (2) is released from the sediments as iron (hydr)oxides there are reduced (Van Cappellen and Ingall, 1994). However, redundant P in seawater due to regeneration under anoxic condition would be balanced by other form of sinks, such as enhanced P burial associated with authigenic calcium phosphate minerals (Van Cappellen and Ingall, 1994).

As for nitrogen, an oxygenated ocean allows for extensive nitrification while inhibiting denitrification, and hence a large nitrate reservoir can build up in seawater (Reinhard et al., 2017). Under anoxic conditions, nitrogen will be preferentially recycled relative to organic carbon, as evidenced by high TOC/TN ratios in the sediments (Twichell et al., 2002). However, N recycling under anoxic condition would be gradually balanced by extensive denitrification.

Therefore, nutrient recycling under anoxic conditions is unlikely to operate permanently. In addition, enhanced P and N burial associated with organic matter could occur under anoxic conditions but normally requires significant absence of oxidants (such as SO_4^{2-}) (Kipp and Stüeken, 2017). This could be the case in the Archean when oxidative weathering was extremely low. However, this is unlikely to have been the case in the Ediacaran, as geochemical data has suggested that SO_4^{2-} contents in the seawater were significantly elevated (Fike et al., 2006; McFadden et al., 2008). Thus, the original sentence “when phosphorus and nitrogen became trapped in the sediments through organic particle deposition-and-burial (line 195-196 in the original manuscript)” has been deleted in the revised manuscript.

Furthermore, the original sentence about elevated loss of nitrate in the sediments in interval V (line 212-215 in the original manuscript) has also been deleted. The low $\delta^{15}\text{N}$ values in interval V may reflect local signatures in the oxygen-depleted zones where N_2 -fixation dominated the nitrogen cycle (-2‰ to 0‰ in the Longbizui and Lijiatuo sections). Please refer to our reply to **Comment #13** and **#15 of Reviewer #3**

Comments #3: (b) The chicken-and-egg problem is not fully resolved, because the termination of anoxic events is explained by increasing bioturbation (l. 211), which would imply a biological trigger. However, how were these bioturbating organisms able to expand while the ocean was still anoxic? A few more sentences would help here.

Revisions/Response: Pulses of oxygenation occurred in the late Ediacaran–early Cambrian, as evidenced by our high $\delta^{15}\text{N}$ in intervals I, III and V and other geochemical data from previous studies (Chen et al., 2015; Scott et al., 2008). The increasing bioturbation was actually facilitated by this increase in oxygen levels.

The increasing bioturbation did not terminate anoxic events, but it did weaken the nutrient regeneration by allowing more oxygen to penetrate into the sediments.

This mechanism helps to stabilize the Earth surface system (line 301-322, please refer to our reply to **Comment #5** of **Reviewer #3**).

On one hand, nutrient regeneration would be gradually balanced by other forms of sinks under continuously anoxic conditions. On the other hand, enhanced marine productivity and burial of organic matter under anoxic condition would lead to increasing oxygen levels in the ocean and atmosphere, eventually counteracting/terminating the ocean anoxia. Thus, on long-term scales (>1 Myr), negative feedbacks would eventually be established, suppressing runaway positive feedbacks between nutrient regeneration, productivity and anoxia. The explanation of this point has been added in the revised manuscript (line 289-300).

Comments #4: (c) The text mentions 'short-term' feedback loops; however, the oxic and anoxic intervals alternate on timescales of several million years as shown in Figure 2. Is it realistic that the build-up of reductants or oxidants and the burial of nutrients would take that long? Thousand-year timescales, corresponding to ocean-mixing timescales, would seem more likely for this proposed mechanism. Perhaps consider external triggers (see next comment).

Revisions/Response: An external force that may have promoted the occurrence of anoxic events during time intervals II, IV and VI, climate cooling, has been added in the revised manuscript.

Nutrients (phosphorus and nitrogen) probably kept being efficiently recycled during the Ediacaran–Cambrian transition, as the seafloor remained largely anoxic due to its being covered by microbial mats (Gehling, 1999). The high nutrient availability could have fueled productivity, promoting local anoxia by creating a positive feedback (Van Cappellen and Ingall, 1994). The key to this positive feedback is the transfer of recycled nutrients from deep waters to the surface ocean, which is closely related to the ocean circulation (Van Cappellen and Ingall, 1994). A recent study of CIA data from South China proposed that a relatively cold and arid climate during Cambrian Stage 2 (interval IV) could have stimulated ocean circulation and encouraged upwelling during the extensive marine transgression (Zhai et al., 2018). It is therefore plausible that climate change may have been an external force promoting episodic anoxia during the Ediacaran–early Cambrian period that bridged the transition from O₂-deficient oceans towards widely oxic oceans. However, CIA data from South China are lacking at the Ediacaran–Cambrian boundary (interval II), and their temporal correlation with our $\delta^{15}\text{N}$ data in Cambrian Stage 3 (intervals V and VI) are uncertain (Zhai et al., 2018). Therefore, more detailed correlation between climate change and redox fluctuations awaits future study.

The explanation of this point has been added and moved to the Section “Dynamic redox fluctuations and nutrient regulation” (line 263-288).

Comments #5: (d) The transition from alternating oxic-anoxic events to a fully oxic ocean (l. 220) is not well explained, because it is not clear why the proposed mechanism would lead to a progression towards more oxic conditions. Is the idea that with the appearance of macro-fauna it would become impossible to go back into an anoxic state? But if so, how would that explain anoxic events in the Phanerozoic? I realize that this discussion may go beyond the scope of the manuscript. Perhaps one possible solution to this and some of the previous points would be an external driver, such as climatic perturbations. See, for example, Zhai et al. 2018 (Paleo3), who proposed global cooling and warming events across the Ediacaran-Cambrian transition as a driver of ocean circulation and productivity.

Revisions/Response: An external force promoting the occurrence of anoxic events in the late Ediacaran–early Cambrian, climate cooling, has been added in the revised manuscript. Please refer to our reply to **Comment #4 of Reviewer #3**.

The high nutrient availability in the ocean could fuel productivity, leading at times to anoxia on continental shelves during the late Ediacaran–early Cambrian. With the evolution of animals and increased bioturbation (Mángano and Buatois, 2014), more oxygen was allowed to penetrate into the sediments, and hence enhanced nutrient (phosphorus) burial in the sediments, providing a key mechanism for regulating nutrient excess and oxygen levels in the Earth system (Boyle et al., 2014). This is also consistent with our relatively lower $\delta^{15}\text{N}$ values in interval V compared to those of intervals I and III (please refer to our reply to **Comment #15 of Reviewer #3**). This might be the reason for environments evolving towards more oxic conditions later in the Phanerozoic. More explanation on this point has been added in the revised manuscript (line 301-322).

On the other hand, animal diversification could help to stabilize the Earth surface system by regulating nutrient cycles. However, it would take a long time, as the so-called “Cambrian substrate revolution” probably lasted for the whole Cambrian Period (Mángano and Buatois, 2017). Therefore, anoxic events could still occur after the appearance of macro-fauna.

Indeed, there were still anoxic events in the Phanerozoic. The causes for these later anoxic events may be different from those in the early Cambrian, because Phanerozoic “mixgrounds” with high bioturbation would not favour nutrient regeneration. Instead, anoxic events in the later Phanerozoic are mainly driven by external forces, such as ocean stagnation, global warming (reduced solubility of oxygen) due to volcanism, changes of sea level and so on (Meyer and Kump, 2008).

Detailed comments:

Comments #6: l. 61: change to 'performed by some prokaryotes'. Otherwise it may give the impression that all prokaryotes fix nitrogen.

Revisions/Response: Combined with **Comment #7 of Reviewer #1**, "prokaryotes" has been revised into "some bacteria and archaea".

Comments #7: ll. 133-157: This proposed linkage between iron speciation and nitrogen isotopes is provocative and an interesting idea. Another possible explanation for the seemingly 'anoxic' iron signature may be that Fe²⁺ upwelled from the anoxic abyssal deep ocean into the Yangtze Basin and got trapped there as authigenic iron as it encountered oxic conditions. This scenario is exemplified by the Peruvian upwelling zone, where authigenic iron is actually slightly depleted within the OMZ but enriched in the oxic sediments immediately underneath (Scholz et al. 2014 GCA). To test this hypothesis it may be worth checking the exact speciation of iron in these sediments.

Revisions/Response: We have checked the exact speciation of iron from seven sections distributed in different sedimentary settings (from shallow to deep waters), and plotted the Fe_{oxides}/Fe_{HR} and Fe_{HR}/Fe_T data against ages (see the figure below, iron speciation data are from Cai et al., 2015; Feng et al., 2014; Och et al., 2013, 2016; Yuan et al., 2014). Neither Fe_{oxides}/Fe_{HR} or Fe_{HR}/Fe_T data show any correlation with our δ¹⁵N data. The values of the Fe_{HR}/Fe_T data suggest that the bottom waters overlying the sediments were dominantly anoxic except for a small number of 'oxic' samples from Yangjiaping section and Xiaotan section (figure below), which means that variation in Fe_{oxides}/Fe_{HR} was likely controlled by redox conditions within the sediments. By contrast, our δ¹⁵N data reflect fluctuations in the redox chemocline in the water column. Thus in cases where the redox chemocline did not reach the sediments, it is to be expected that iron speciation does not co-vary with the δ¹⁵N data.

Therefore, these results are consistent with our conclusion that pulses of oxygenation in deep waters occurred during the late Ediacaran–early Cambrian, with anoxia restricted to near, or even at the sediment-water interface.

Comments #8: ll. 164-166: Note that P bioavailability also depends on the supply of oxidants for remineralization of organic matter (Kipp & Stüeken 2017 Science Advances). It is not certain that enhanced P delivery by weathering could on its own act as a trigger for enhanced N₂ fixation. I would suggest rephrasing this part of the text to account for these uncertainties.

Revisions/Response: Revised as suggested. Other mechanisms for increased phosphorus contents have been considered and added in the revised manuscript.

The original sentence has been revised to “Increasing seawater phosphate could have been due to enhanced weathering, following major continental reorganisation and Cryogenian deglaciation³⁹, an elevated supply of oxidants for organic remineralization⁴⁰, and/or decreased scavenging through co-precipitation of Fe and P⁴¹” (line 195-200)

Comments #9: l. 169: The build-up of this nitrate reservoir would also have

required the establishment of oxic conditions. That should be stressed here.

Revisions/Response: Revised as suggested.

The original sentence has been revised to “a stable reservoir of nitrate existed in the substantially oxygenated ocean of the late Ediacaran–early Cambrian, as revealed by our $\delta^{15}\text{N}$ data for the time intervals I, III and V” (line 206-208)

Comments #10: l. 171-172: It’s not clear why eukaryotic phytoplankton would have had a competitive advantage over N_2 -fixing prokaryotes – they would still have depended on them. It should be sufficient to say that eukaryotic phytoplankton were able to thrive under these conditions. Perhaps cite Brocks et al. 2017 (Nature), who provide biomarker evidence for the radiation of eukaryotic algae at this time.

Revisions/Response: Revised as suggested.

Planktonic N -fixing cyanobacteria have been suggested to radiate into the open ocean during the late Neoproterozoic, substantially changing nitrogen supply in seawater (Sánchez-Baracaldo et al., 2014; Sánchez-Baracaldo, 2015). Enhanced nutrient supply therefore modified the ecological structure of primary producers, likely facilitating a shift in phytoplankton composition from small-celled, cyanobacterial picoplankton towards large-celled, planktonic algae in the Neoproterozoic as evidenced from biomarker and molecular clock data (Brocks et al., 2017; Sanchez-Baracaldo et al, 2017). More explanation of this point has been added in the revised manuscript (line 190-205).

The original sentence “Thus eukaryotic phytoplankton could have enjoyed a competitive advantage versus nitrogen-fixing prokaryotes” has been revised to “Large-celled, eukaryotic phytoplankton preferentially assimilate nitrate over ammonia⁴⁴, and so were able to thrive under nitrate-replete conditions, as diffusion limitation of nutrient uptake increases with cell size⁴⁵” (line 208-211). Please also see our reply to **Comment #12** of **Reviewer #2**.

Comments #11: l. 194: Delete ‘fortunately’. That’s a bit of a subjective term.

Revisions/Response: Revised as suggested.

Comments #12: ll. 201-221: The writing style is noticeably different in this part of the manuscript. I suggest rephrasing some of the sentences for clarity.

Revisions/Response: Revised as suggested (see line 263-322 in the revised manuscript).

Comments #13: l. 214: It is true that a nitrogen cycle dominated by sedimentary denitrification would show $\delta^{15}\text{N}$ values around 0 permil (Quan & Falkowski 2009 Geobiology, Figure 4). However, in an ocean that has oxygen minimum zones somewhere, this is an unlikely scenario. Even the modern ocean shows the effects of water-column denitrification in locally suboxic regions. These light $\delta^{15}\text{N}$ values are more parsimoniously interpreted as an N_2 -fixation dominated nitrogen cycle.

Revisions/Response: Revised as suggested.

It is true that the low $\delta^{15}\text{N}$ values in interval V more likely reflect local signatures in the oxygen-depleted zones where N_2 -fixation dominated the nitrogen cycle (-2‰ to 0‰ in the Longbizui and Lijiatuo sections, Fig. 2d). This has been revised in the manuscript (line 301-308). Please also refer to our reply to **Comment #15** of **Reviewer #3**.

Comments #14: l. 247-257: This section on correlative data from Kazakhstan is a strength of the paper. I would suggest moving it further up, closer to l. 222 where the global significance of the data is first discussed.

Revisions/Response: Revised as suggested.

Comments #15: l. 258-259: The text mentions a trend here in $\delta^{15}\text{N}$ that indicates progressive oxygenation into the Cambrian. However, in Figure 2, the Cambrian $\delta^{15}\text{N}$ values are actually similar or even lighter than $\delta^{15}\text{N}$ values in the Ediacaran. This needs to be reconciled with the inferred oxygenation. I'm wondering if some of the light $\delta^{15}\text{N}$ data in the Cambrian sections reflect local paleogeography (restricted basins). Is that possible?

Revisions/Response: The average $\delta^{15}\text{N}$ value of interval V is indeed relatively compared to those of intervals I and III (Fig. 2d). We do not exclude the possibility that some of the low $\delta^{15}\text{N}$ values in interval V may reflect local signatures in oxygen-depleted zones (-2‰ to 0‰ in the Longbizui and Lijiatuo sections, Fig. 2d), as the sea level may have dropped following the extensive transgression during Cambrian Stage 2 (Feng et al., 2014; Wang et al., 2015). However, it is more likely that lower $\delta^{15}\text{N}$ values represent global ocean deoxygenation in interval V, as proposed previously (Boyle et al., 2014; Xiang et al., 2017). Increased bioturbation could have enhanced nutrient burial (phosphorus) in the sediments, leading to a decrease in the global oxygen reservoir relative to interval I and III (Boyle et al., 2014). The detailed explanation has been added in the Section "Dynamic redox fluctuations and nutrient regulation" (line 301-322).

Here the original phrase "the trend towards greater oxygenation" has been revised into "pulses of oxygenation" (line 381).

Comments #16: Lastly, the authors mention phosphate deposition in the basin (ll. 84-85). Are there temporal or spatial trends in phosphate abundances that could be integrated into the discussion?

Revisions/Response: Our study shows that nitrogen and carbon isotopic values exhibit positive excursions in intervals I, III and V, and negative excursions in intervals II, IV and VI during late Ediacaran–early Cambrian (Fig. 2). However, during Ediacaran–early Cambrian phosphorites were deposited only in two episodes over the shallow-water carbonate platform in South China. The first episode corresponds to deposition of the Doushantuo Formation of the early Ediacaran (ca. 635 Ma to 551 Ma), which is prior to the time interval of our study.

The second episode is in Cambrian Fortunian Stage, namely interval III of our study (see supplementary Fig. 3). It seems that the temporal occurrence of phosphorite deposits were much less frequent than the C-, N- isotopic variations during the studied period. The possible reason is that formation of phosphorite deposits was complex, and associated with multiple factors, such as marine redox state, phosphorus reservoir, ocean currents, biological processes, basin geology, local upwelling and so on (Pufahl and Hiatt, 2011).

I hope, these comments will be helpful with the publication of the manuscript.

Thanks!

Best regards,
Eva Stüeken

REVIEWERS' COMMENTS:

Reviewer #1 (Remarks to the Author):

The authors have satisfactorily addressed all of my original concerns and comments, and have submitted a stronger and more nuanced manuscript. Despite a few more minor recommendations listed below, I would be happy to see this paper published in Nature Communications.

Specific Comments:

L 44-45: This is primarily referring to the Ediacara biota, right? Might want to add 'in the fossil record' then after 'appeared,' since animals, and other macroscopic multicellular eukaryotes (e.g. red algae), most likely evolved before the Ediacaran.

L 214: The trend towards more abundant eukaryotic phytoplankton?

L 225: Replace 'force' with 'selective driver' or 'selective pressure' (you can replace 'pressure' on L 223 with 'activity' to avoid overusing the word).

L 288: According to Butterfield's argument, wasn't the initial trigger the evolution of animals themselves, which was primarily dictated by genetic factors and evolutionary contingency, and independent of oxygenation?

Refs:

Butterfield, N. J. Proterozoic photosynthesis--a critical review. *Palaeontology* 58, 953–972 (2015).

Butterfield, N. J. Early evolution of the Eukaryota. *Palaeontology* 58, 5–17 (2015).

L 229: I'm confused – sponges are often considered suspension feeders (e.g. Gili & Coma 1998 *Trends in Ecology & Evolution*). What animal taxa are you referring to here? Suspension-feeding arthropods? And by 'appear,' do you specifically mean in the fossil record?

L 232: Maybe say 'which would have placed a selective pressure...'

L 300: Could you be more specific? What kinds of life do you mean?

Reviewer #2 (Remarks to the Author):

I have gone over the revised manuscript, and I am pleased with the changes and improvement the authors have made. The authors data set add valuable insights into the Nitrogen biochemical cycle at the end of the Pre-Cambrian.

I look forward to seeing this manuscript published in Nature Communications.

Reviewer #3 (Remarks to the Author):

Review of: Wang et al., resubmission

The authors have made significant changes to the previous version of the manuscript and

addressed all major issues. The text is well-written and will be a valuable contribution to the community. I only have a few additional comments for clarification.

Main comment:

- It is important to note that these anoxic intervals are not comparable in $\delta^{15}\text{N}$ to earlier Proterozoic settings, where $\delta^{15}\text{N}$ rarely drops below -1 permil (reviewed by Stüeken et al. 2016 Earth Science Reviews). Something was qualitatively different about this time, which would be worth stressing more explicitly. It is otherwise not clear why there weren't more frequent oxygenation intervals throughout the earlier Precambrian as a result of the feedback loops described in the text.

- Until the end of the discussion where the global significance of the data is addressed more explicitly, it is unclear from the text if the described trends are indeed reflecting global redox changes rather than basinal oceanographic events. I suggest mentioning earlier in the introduction that additional data from Kazakhstan were included in this study. Ideally, these data could be integrated directly into the discussion of the Yangtze Basin. Furthermore, the description of the geologic setting would be strengthened with a few statements about the connectivity of the basin to the open ocean – to the extent that this is known. The sentences about phosphorite deposition can probably be removed, because phosphorites are not discussed again later in the text.

Line comments (referring to the marked copy of the manuscript):

I. 71: It would help to add in parentheses the numerical value of the fractionation imparted by biological N_2 fixation.

I. 121: Maybe change this sentence to 'all nitrogen isotopes curves from the different lithofacies...' or something like that. Otherwise it is not clear from the text what the curves are.

I. 129: When mentioning 'significant difference(s)', such as here, it would strengthen the argument if p-values could be quoted, or if the box & whisker plot from the response to the previous comments could be included in the supplementary material.

II. 168-169: The absolute value of $\delta^{15}\text{N}$ values cannot be directly correlated with dissolved oxygen levels. There are many competing factors (relative abundance of N_2 -fixers in sediments; the magnitude of the fractionation during denitrification; the nitrate demand in the photic zone; diagenetic alteration; metamorphism). Without quantitative assessments of these parameters, $\delta^{15}\text{N}$ is mostly an 'on/off' proxy for the presence of an aerobic nitrogen cycle. I would therefore recommend removing this statement about relatively lower O_2 levels than today.

I. 244: I would suggest removing the N_2 subscript from $\delta^{15}\text{N}$, because it's not used elsewhere in the text until now.

II. 245-247: You may want to consider the possibility of biological N_2 -fixation in the presence of high iron levels (Zerkle et al. 2008 JGR, Zhang et al. 2014 PNAS) to explain these light $\delta^{15}\text{N}$ values.

I. 259: Is there evidence from iron speciation that these deep waters were rich in H_2S ? If yes, that should be mentioned here.

II. 265-267: Why would P and N be efficiently recycled under anoxic conditions? This is unclear, especially for N.

I. 269: Is there evidence of high productivity during the anoxic events? Maybe elevated TOC or

other indicators that can be quoted?

I. 271: What is 'superfine' organic matter? Can this be quantified as a size fraction?

I. 274: Why would anoxia boost nutrient regeneration in a positive feedback? Ammonium is only known to build up in restricted, density-stratified basins today, such as the Black Sea, the Cariaco Basin or stratified lakes. Nitrate is lost under anoxic conditions. Note that the earlier Precambrian anoxic oceans are thought to be nutrient-poor.

I. 281-284: If enhanced ocean circulation brings up nutrients, why would that trigger oxygenation? The opposite was argued in II. 268-272.

II. 289-292: This sentence contradicts itself. Are nutrients recycled (I. 289) or lost (II. 291-292)?

Fig. 3: NH_3 should be NH_4^+ (the pKa is roughly 9.2 at standard conditions). The second panel should make it clear that the inferred anoxic conditions only existed regionally, as discussed in the main text. Perhaps it would be better to draw an expanded ODZ with oxic waters dominating offshore.

Supplements:

II. 45-47: Note that kerogen $\delta^{15}\text{N}$ does in fact change during metamorphic alteration. It appears to become lighter initially at low metamorphic grade, but heavier in amphibolite facies where N-loss can be substantial (Stüeken et al. 2017 GCA).

Best wishes,
Eva Stüeken

The line numbers mentioned below all refer to the marked manuscript.

Replies to the Reviewer #1

Reviewer #1 (Remarks to the Author):

The authors have satisfactorily addressed all of my original concerns and comments, and have submitted a stronger and more nuanced manuscript. Despite a few more minor recommendations listed below, I would be happy to see this paper published in Nature Communications.

Thanks for your constructive comments and suggestions! .

Specific Comments:

Comments #1: L 44-45: This is primarily referring to the Ediacara biota, right? Might want to add 'in the fossil record' then after 'appeared,' since animals, and other macroscopic multicellular eukaryotes (e.g. red algae), most likely evolved before the Ediacaran.

Revisions/Response: Revised as suggested.

This sentence has been revised to "Soft-bodied macroscopic multicellular Ediacara organisms, including animals, appeared abruptly in the fossil records during the late Ediacaran Period¹" (line 39-40).

Comments #2: L 214: The trend towards more abundant eukaryotic phytoplankton?

Revisions/Response: Here we did refer to "the trend towards more abundant eukaryotic phytoplankton", and have now revised it as suggested (line 183-184).

Comments #3: L 225: Replace 'force' with 'selective driver' or 'selective pressure' (you can replace 'pressure' on L 223 with 'activity' to avoid overusing the word).

Revisions/Response: Revised as suggested.

Comments #4: L 288: According to Butterfield's argument, wasn't the initial trigger the evolution of animals themselves, which was primarily dictated by genetic factors and evolutionary contingency, and independent of oxygenation?

Revisions/Response: There are now increasing arguments about the causal relationship between environmental changes and biological innovation. The conventional view is that oxygenation was the main trigger behind the animal evolution (Knoll, 2003; Nursall, 1959). Genetic factors could be one possible initial trigger for animal evolution (Butterfield, 2015a, 2015b), as some of the most primitive animals (such as sponges) could have survived in a low oxygen environment (Mills et al., 2014a; Sperling et al., 2015). Other factors, such as nutrients and climate, have also been previously proposed (Mills et al., 2014b; Shields et al., 2017).

Therefore, the sentence here has been revised to “This may lead to a chicken-and-egg question, as the initial trigger for the positive feedback described above has still been in debate” (line 196-197).

Refs:

Butterfield, N. J. Proterozoic photosynthesis--a critical review. *Palaeontology* 58, 953–972 (2015a).

Butterfield, N. J. Early evolution of the Eukaryota. *Palaeontology* 58, 5–17 (2015b).

Mills, D. B, et al. Oxygen requirements of the earliest animals. *Proceedings of the National Academy of Sciences* 111, 4168-4172 (2014a).

Mills, D. B, Canfield, D. E. Oxygen and animal evolution: Did a rise of atmospheric oxygen “trigger” the origin of animals? *BioEssays* 36, 1145-1155 (2014b).

Comments #5: L 229: I’m confused – sponges are often considered suspension feeders (e.g. Gili & Coma 1998 *Trends in Ecology & Evolution*). What animal taxa are you referring to here? Suspension-feeding arthropods? And by ‘appear,’ do you specifically mean in the fossil record?

Revisions/Response: Sponges are indeed suspension feeders. To make it clear, we has revised “suspension-feeding animals” to “suspension-feeding mesozooplankton”. Here “mesozooplankton” refers to some suspension-feeding arthropods, and was preserved in the fossil record in the early Cambrian (Harvey et al., 1998).

The sentence has been revised to “The earliest fossil record for suspension-feeding mesozooplankton was not reported to appear until the early Cambrian (510–515 Ma)⁴⁸” (line 197-199)

Comments #6: L 232: Maybe say ‘which would have placed a selective pressure...’

Revisions/Response: Revised as suggested.

Comments #7: L 300: Could you be more specific? What kinds of life do you mean?

Revisions/Response: The word “early life” has been revised to “metazoans” who requirement oxygen for aerobic metabolism (line 272)..

Replies to the Reviewer #2

Reviewer #2 (Remarks to the Author):

I have gone over the revised manuscript, and I am pleased with the changes and improvement the authors have made. The authors data set add valuable insights into the Nitrogen biochemical cycle at the end of the Pre-Cambrian.

I look forward to seeing this manuscript published in Nature Communications.
Thanks for your constructive and supportive comments!

Replies to the Reviewer #3

Reviewer #3 (Remarks to the Author):

Review of: Wang et al., resubmission

The authors have made significant changes to the previous version of the manuscript and addressed all major issues. The text is well-written and will be a valuable contribution to the community. I only have a few additional comments for clarification.

Thanks for your constructive comments and suggestions! .

Main comment:

Comments #1: It is important to note that these anoxic intervals are not comparable in $\delta^{15}\text{N}$ to earlier Proterozoic settings, where $\delta^{15}\text{N}$ rarely drops below -1 permil (reviewed by Stüeken et al. 2016 Earth Science Reviews). Something was qualitatively different about this time, which would be worth stressing more explicitly. It is otherwise not clear why there weren't more frequent oxygenation intervals throughout the earlier Precambrian as a result of the feedback loops described in the text.

Revisions/Response: Revised as suggested.

Low \$\delta^{15}\text{N}\$ values below \$-2\text{‰}\$ occurred in intervals II, IV and VI, indicating partial \$\text{NH}_4^+\$ -assimilation with a large isotopic fractionation effect (Higgins et al., 2012). These low \$\delta^{15}\text{N}\$ values (\$< -2\text{‰}\$ ) in anoxic intervals II, IV and VI rarely occurred in earlier Proterozoic (Stüeken et al. 2016), implying that marine nitrogen reservoir during the late Ediacaran-early Cambrian may be qualitatively different from the earlier times, and a larger \$\text{NH}_4^+\$ reservoir has been built up in the anoxic waters. The explanation of this point has been added in the revised manuscript (line 218-224).

Comments #2: Until the end of the discussion where the global significance of the data is addressed more explicitly, it is unclear from the text if the described trends are indeed reflecting global redox changes rather than basinal oceanographic events. I suggest mentioning earlier in the introduction that additional data from Kazakhstan were included in this study. Ideally, these data could be integrated directly into the discussion of the Yangtze Basin. Furthermore, the description of the geologic setting would be strengthened with a few statements about the connectivity of the basin to the open ocean – to the extent that this is known. The sentences about phosphorite deposition can probably be removed, because phosphorites are not discussed again later in the

text.

Revisions/Response: Revised as suggested.

We have now added a sentence about global significance of redox fluctuations in the introduction (line 85-87). However, $\delta^{15}\text{N}$ data from Kazakhstan has not been included here, as global significance of redox fluctuations has been supported by coupling of $\delta^{15}\text{N}$ and $\delta^{13}\text{C}$ data in South China, $\delta^{15}\text{N}$ data from Kazakhstan and some fossil records in South China, Siberia, Mongolia and Morocco.

The description on the connectivity of the basin to the open ocean has been added in the revised manuscript (line 90-92): “The Yangtze block gradually evolved from a rift basin to a passive continental margin during the Proterozoic–Cambrian transition, with a southwest facing connection to the open ocean²³”.

The sentences about phosphorite deposition has been revised to “During this time interval, carbonate predominantly deposited over the shallow-water platform, before migrating basinward into the black muds (i.e. cherts and shales) of the deeper marine environment²⁴ (Fig. 1b)” (line 94-96)

Line comments (referring to the marked copy of the manuscript):

Comments #3: l. 71: It would help to add in parentheses the numerical value of the fractionation imparted by biological N_2 fixation.

Revisions/Response: Revised as suggested.

“Nitrogen enters the oceanic N cycle mainly through the fixation of atmospheric N_2 (N_2 -fixation) by autotrophs using Mo-nitrogenases with minor isotopic fractionation ($\leq 2\text{‰}$)” (line 63-65).

Comments #4: l. 121: Maye change this sentence to ‘all nitrogen isotopes curves from the different lithofacies...’ or something like that. Otherwise it is not clear from the text what the curves are.

Revisions/Response: Revised as suggested.

The sentence has been revised to “All $\delta^{15}\text{N}$ curves from the different lithofacies show similar trends” (line 115).

Comments #5: l. 129: When mentioning ‘significant difference(s)’, such as here, it would strengthen the argument if p-values could be quoted, or if the box & whisker plot from the response to the previous comments could be included in the supplementary material.

Revisions/Response: Revised as suggested.

The box & whisker plot from the response to the previous comments has now been included in the supplementary material (Supplementary Note 4 and Supplementary Fig. 6).

Comments #6: ll. 168-169: The absolute value of $\delta^{15}\text{N}$ values cannot be directly

correlated with dissolved oxygen levels. There are many competing factors (relative abundance of N₂-fixers in sediments; the magnitude of the fractionation during denitrification; the nitrate demand in the photic zone; diagenetic alteration; metamorphism). Without quantitative assessments of these parameters, $\delta^{15}\text{N}$ is mostly an 'on/off' proxy for the presence of an aerobic nitrogen cycle. I would therefore recommend removing this statement about relatively lower O₂ levels than today.

Revisions/Response: Revised as suggested. The sentence "although the lower mean $\delta^{15}\text{N}$ value suggests lower NO₃⁻ concentrations than in the modern ocean" has now been deleted in the revised manuscript (line 155-156).

Comments #7: l. 244: I would suggest removing the N₂ subscript from $\delta^{15}\text{N}$, because it's not used elsewhere in the text until now.

Revisions/Response: Revised as suggested.

Comments #8: ll. 245-247: You may want to consider the possibility of biological N₂-fixation in the presence of high iron levels (Zerkle et al. 2008 JGR, Zhang et al. 2014 PNAS) to explain these light $\delta^{15}\text{N}$ values.

Revisions/Response: Revised as suggested. The possibility of N₂-fixation using Fe-nitrogenases has been considered in the revised manuscript (line 214-218).

"The very low $\delta^{15}\text{N}$ values (< -2‰) could be interpreted as N₂-fixation using Fe-nitrogenases with a large isotopic fractionation (~8‰, ref. 49). However, Fe-nitrogenases were generally expressed only when Mo availability in seawater is extremely low⁴⁹, which is not the case in the Ediacaran-early Cambrian with high Mo concentrations (>100 ppm) recorded in the sediments^{11, 13, 14}."

Comments #9: l. 259: Is there evidence from iron speciation that these deep waters were rich in H₂S? If yes, that should be mentioned here.

Revisions/Response: Some iron speciation data did suggest that anoxic deep waters were rich in H₂S in South China basin (Feng et al., 2014; Och, et al., 2016; Yuan et al., 2014). Here the sentence has been revised to "The spread of anoxic waters containing toxic H₂S, as evidenced by iron speciation data^{33, 51, 52}, may have led to the extinction of Ediacaran- and some small-shelly type animals (Fig. 2a)" (line 230-231)

Comments #10: ll. 265-267: Why would P and N be efficiently recycled under anoxic conditions? This is unclear, especially for N.

Revisions/Response:

Under oxic conditions, phosphorus (phosphate) will be co-precipitated with iron (hydr)oxides or adsorbed onto their surfaces, and hence retained in the sediments (Filippelli, 2008). Under anoxic conditions, phosphorus (1) is increasingly lost from organic matter (Ingall et al., 2005); and (2) is released from the sediments as iron (hydr)oxides there are reduced (Van Cappellen and Ingall, 1994). Therefore, phosphorus would be more efficiently recycled under

anoxic conditions.

As for nitrogen, it will be preferentially recycled relative to organic carbon, as nitrogen-rich proteins are usually preferentially degraded than nitrogen-poor carbohydrates (Meyers and Arnaboldi, 2008). Meanwhile, protein degradation, and hence N-recycling efficiency, may be independent of water column redox state (Pantoja et al., 2004). Thus we have deleted the speculation of efficient N-recycling under anoxic condition in the revised manuscript.

Furthermore, our $\delta^{15}\text{N}$ data has suggested that a large fixed nitrogen reservoir has been built up in seawater during the late Ediacaran–early Cambrian, in form of either NO_3^- under oxic conditions in intervals I, III and V, or NH_4^+ under anoxic conditions in intervals II, IV and VI.

Therefore, the sentence here has been revised to “A large fixed nitrogen reservoir has been built up in seawater, making nitrogen no longer a limiting element during the late Ediacaran–early Cambrian. The bio-limiting nutrient phosphorus probably kept being efficiently recycled from organic matter and iron (hydr)oxides during the Ediacaran–Cambrian transition, as the seafloor remained largely anoxic due to its being covered by microbial mats^{52, 53}” (line 235-240).

Comments #11: l. 269: Is there evidence of high productivity during the anoxic events? Maybe elevated TOC or other indicators that can be quoted?

Revisions/Response: TOC contents in interval IV are high at all the sections in our study (Cremonese et al., 2013, 2014; Cai et al., 2015; Wang et al., 2015; Zhang et al., 2017), probably reflecting high productivity. However, TOC contents in intervals II and VI do not exhibit increasing trend at most of the studied sections, such as Yanjia section (this study), Longbizui section (Cremonese et al., 2014), Sancha section (Wang et al., 2015). The possible reason is that TOC contents in the sediments are associated with many factors, including productivity, lithology, sedimentary rate, remineralization etc. Thus evidence for high productivity during anoxic events await further study.

Comments #12: l. 271: What is ‘superfine’ organic matter? Can this be quantified as a size fraction?

Revisions/Response: The word “superfine” here roughly refers to smaller-sized organic matter which sinks through the water column relatively slower. We do not know the exact size. To make it less confusing, we have deleted “superfine” in the revised manuscript (line 243).

Comments #13: l. 274: Why would anoxia boost nutrient regeneration in a positive feedback? Ammonium is only known to build up in restricted, density-stratified basins today, such as the Black Sea, the Cariaco Basin or

stratified lakes. Nitrate is lost under anoxic conditions. Note that the earlier Precambrian anoxic oceans are thought to be nutrient-poor.

Revisions/Response: The very low $\delta^{15}\text{N}$ values ($< -2\text{‰}$) in intervals II, IV and VI could have resulted from partial NH_4^+ -assimilation with a large isotopic fractionation effect (Higgins et al., 2012). These low $\delta^{15}\text{N}$ values ($< -2\text{‰}$) rarely occurred in earlier Proterozoic (Stüeken et al. 2016), implying that marine nitrogen reservoir during the late Ediacaran–early Cambrian may be qualitatively different from the earlier times, and a large NH_4^+ reservoir has built up in the anoxic waters. Please also refer to our reply to **Comments #1 of Reviewer #3**.

Meanwhile, the bio-limiting nutrient phosphorus was more efficiently recycled from organic matter and iron (hydr)oxides under anoxic conditions. Please also refer to our reply to **Comments #10 of Reviewer #3**

Therefore, if climate cooling stimulated ocean circulation, upwelling waters would bring up abundant nutrients, including P and N, from the deep water to the surface water. The high nutrient availability boosted marine productivity, enhancing deep-water anoxia, which in turn encouraged benthic phosphorus regeneration. Thus anoxia actually enhanced phosphorus regeneration in a positive feedback.

The sentence here has been revised to “which in turn boosted nutrient (phosphorus) regeneration in a process of positive feedback” (line 245).

Comments #14: l. 281-284: If enhanced ocean circulation brings up nutrients, why would that trigger oxygenation? The opposite was argued in ll. 268-272.

Revisions/Response: Enhanced ocean circulation promoted the anoxia in intervals II, IV and VI, rather than triggering ocean oxygenation in intervals I, III and V. Instead, increased nutrient (such as nitrogen) availability facilitated ocean oxygenation during the late Ediacaran–early Cambrian transition, by boosting the primary productivity of large-celled, eukaryotic phytoplankton in the ocean.

Actually, enhanced ocean circulation could result in different consequences, as the upwelling deep waters were dominantly anoxic in the Precambrian, but oxic in the later Phanerozoic. The Ediacaran to early Cambrian interval bridges the transition from O_2 -deficient oceans to widely oxic oceans. The upwelling deep waters were probably O_2 -deficient, and contain abundant nutrients (N and P) during the Ediacaran–early Cambrian period. Therefore, we emphasized that “climate change may have been an external force promoting episodic anoxia during the Ediacaran–early Cambrian period that bridged the transition from O_2 -deficient oceans towards widely oxic oceans” (line 251-254).

Comments #15: ll. 289-292: This sentence contradicts itself. Are nutrients recycled (l. 289) or lost (ll. 291-292)?

Revisions/Response: Phosphorus was efficiently recycled under anoxic conditions. However, when phosphorus concentrations was high enough to approach the threshold of saturation, phosphorus would deposit in form of calcium phosphate minerals in the sediments (Van Cappellen and Ingall, 1994). N-recycling efficiency is independent of water column redox state (Pantoja et al., 2004), and nitrogen would be increasingly lost through extensive denitrification and/or anammox in O₂-deficient environments.

Therefore, the sentence here has been revised to “Although phosphorus was more efficiently recycled from sediments under continuously anoxic conditions, increases in other P sinks, such as authigenic calcium phosphate, would eventually balance out sources and sinks⁵⁶. Nitrogen enrichment in seawater, as revealed by the very low $\delta^{15}\text{N}$ values in intervals II, IV and VI, would also be balanced through enhanced denitrification (and/or anammox) in O₂-deficient environments¹⁹” (line 258-265)

Comments #16: Fig. 3: NH₃ should be NH₄⁺ (the pKa is roughly 9.2 at standard conditions). The second panel should make it clear that the inferred anoxic conditions only existed regionally, as discussed in the main text. Perhaps it would be better to draw an expanded ODZ with oxic waters dominating offshore.

Revisions/Response: Revised as suggested.

Supplements:

Comments #17: ll. 45-47: Note that kerogen $\delta^{15}\text{N}$ does in fact change during metamorphic alteration. At appears to become lighter initially at low metamorphic grade, but heavier in amphibolite facies where N-loss can be substantial (Stüeken et al. 2017 GCA).

Revisions/Response: Revised as suggested.

The sentence has been revised to “Metamorphism could increase $\delta^{15}\text{N}$ values in phyllosilicates while $\delta^{15}\text{N}$ values in organic matter become lighter¹⁰. However, the isotopic effect of nitrogen loss from the system has been suggested to be ~1-2‰ units at greenschist facies and ~3-4‰ units at amphibolite facies^{10,11}” (line 45-49 in the revised Supplementary Information).

Thanks again !

Best wishes,
Eva Stüeken